# Biogenesis of a bacterial metabolosome for propanediol utilization

Mengru Yang[1], Nicolas Wenner[2], Gregory F. Dykes[1], Yan Li[2], Xiaojun Zhu[2], Yaqi Sun[1], Fang Huang[1], Jay C. D. Hinton [2] & Lu-Ning Liu [1,3✉]

Bacterial metabolosomes are a family of protein organelles in bacteria. Elucidating how thousands of proteins self-assemble to form functional metabolosomes is essential for understanding their significance in cellular metabolism and pathogenesis. Here we investigate the de novo biogenesis of propanediol-utilization (Pdu) metabolosomes and characterize the roles of the key constituents in generation and intracellular positioning of functional metabolosomes. Our results demonstrate that the Pdu metabolosome undertakes both "Shell first" and "Cargo first" assembly pathways, unlike the β-carboxysome structural analog which only involves the "Cargo first" strategy. Shell and cargo assemblies occur independently at the cell poles. The internal cargo core is formed through the ordered assembly of multiple enzyme complexes, and exhibits liquid-like properties within the metabolosome architecture. Our findings provide mechanistic insight into the molecular principles driving bacterial metabolosome assembly and expand our understanding of liquid-like organelle biogenesis.

[1] Institute of Systems, Molecular and Integrative Biology, University of Liverpool, Crown Street, Liverpool L69 7ZB, United Kingdom. [2] Institute of Infection, Veterinary & Ecological Sciences, University of Liverpool, Crown Street, Liverpool L69 7ZB, United Kingdom. [3] College of Marine Life Sciences, and Frontiers Science Center for Deep Ocean Multispheres and Earth System, Ocean University of China, Qingdao 266003, China. ✉email: luning.liu@liverpool.ac.uk

Compartmentalization of metabolic pathways within eukaryotic and prokaryotic cells both enhances and regulates energy and metabolism[1]. Unlike eukaryotes that employ lipid-bound organelles to perform specific metabolic functions, many bacteria have evolved proteinaceous organelles, termed bacterial microcompartments (BMCs), that allow key metabolic processes to function in the cytoplasm of bacterial cells[2–4]. BMCs consist of a single-layer proteinaceous shell that encapsulates both enzymes and metabolites. The virus-like polyhedral shell serves as a physical barrier to control the passage of metabolites and facilitates catabolism in a sequestered micro-environment[5–8]. The highly evolved structural and assembly features of BMCs enable the special bacterial organelles to play pivotal roles in $CO_2$ fixation, infection, and microbial ecology across diverse bacterial species[2,9,10].

The best-characterized BMCs are the carboxysomes (anabolic BMCs, including α- and β-carboxysomes based on the types of Rubisco encapsulated), which enhance $CO_2$ fixation in all cyanobacteria and many chemoautotrophs[11–14]. In contrast, the 1,2-propanediol (1,2-PD) utilization microcompartment (Pdu BMC) is a model for catabolic BMCs or metabolosomes. A critical stage of Salmonella pathogenesis is the colonization of the mammalian gastrointestinal tract, which requires the catabolism of a number of carbon sources, including 1,2-PD[15]. The Pdu BMC plays a vital role in the breakdown of 1,2-PD during gastrointestinal pathogenesis and promotes bacterial fitness in specific anoxic environmental niches[16–22].

The architecture of Pdu BMCs is constructed by more than ten thousand protein subunits of 18 to 20 different types via intrinsic protein self-assembly[23–26]. Based on current knowledge, the Pdu BMC shell in the pathogenic bacterium Salmonella enterica serovar Typhimurium (S. Typhimurium) contains nine types of protein paralogs: PduA, PduB, PduB′, PduM, PduN, PduJ, PduK, PduT, and PduU[27,28]. The hexameric protein PduJ is the most abundant shell protein within the Pdu BMC in S. Typhimurium LT2, followed by PduA and PduBB′[24]. PduB and PduB' are encoded by overlapping genes, and PduB has an N-terminal 37 amino acid extension that is absent from PduB′[29]. The pentameric protein PduN occupies the vertices of the polyhedral shell[30]. PduBB′, PduA or PduJ, and PduN are the shell components that are essential for Pdu BMC formation and structure[27,31]. PduM is a structural protein, which is much less abundant than PduN[24] and has an unknown function within the Pdu BMC[28]. PduK is another minor shell protein within the Pdu BMC[24].

The enzymes involved in 1,2-PD degradation within the Pdu BMC comprise PduCDE (diol dehydratase)[29], PduL[32], PduP[33], PduQ[34], and PduW[35]. Additional enzymes required for reactivation of diol dehydratase and vitamin $B_{12}$ recycling are PduS[36], PduO[37], PduGH[16], and PduX (L-threonine kinase)[38]. PduV was proposed to connect with the filament-associated movement of BMCs within the cell[39]. Previous studies illustrated that the short N-terminal sequences of PduD, PduP, and PduL function as encapsulation peptides and are important for directing enzymes into the Pdu BMC[40–42]. The PduB N-terminus has been reported to bind the shell to cargo enzymes[43].

For decades, it has remained unclear how thousands of protein components self-recognize and assemble in the highly dynamic cellular cytoplasm to generate intact, functional BMCs. Our knowledge of BMCs is largely based on studies on β-carboxysome biogenesis in cyanobacteria. Assembly of the anabolic β-carboxysomes adopts an "Inside out" or "Cargo first" pathway with the assistance of chaperones: Rubisco enzymes first assemble, by interacting with the linker protein CcmM, to form a liquid-like matrix, and then the Rubisco condensate is encapsulated by shell proteins to form a defined carboxysome structure[44–48]. The architecture and protein organization of functional β-carboxysomes exhibit variability in response to changing environments[49]. Unlike the assembly pathway of β-carboxysomes, α-carboxysome assembly appears to follow either the "Shell first" or "Concomitant shell–core assembly" routes: the shell assembles alone without Rubisco packing[50] or concomitantly with nucleation of Rubisco[51,52]. Consistently, empty α-carboxysome shells have been reconstituted in heterologous hosts in the absence of cargo enzymes[53]. In contrast, little is known about the assembly principles that mediate the formation of bacterial metabolosomes, which are required for the catabolism of various substrates and are associated with numerous human diseases.

Here, we design a system to systematically investigate the structural roles of individual Pdu protein components and characterize the spatiotemporal biogenesis of Pdu metabolosomes in S. Typhimurium LT2. Genetic analysis, live-cell fluorescence imaging, electron microscopy (EM), and growth assays reveal that the Pdu components undergo independent self-assembly to form the cargo and shell aggregates. The biogenesis of Pdu metabolosomes adopts a combination of the "Shell first" and "Cargo first" pathways and occurs at the cell poles. We show how the PduB N-terminus and PduM mediate the physical binding between the shell and internal enzyme assemblies, and how PduK impacts the subcellular positioning of Pdu metabolosomes. We also find that the ordered assembly of cargo enzymes generates an internal enzymatic core that possesses a liquid-like status, a physical property that is fundamental for the phase transitions of multicomponent systems. Our findings provide mechanistic insight into the assembly principles of Pdu BMCs, which can be extrapolated to a range of bacterial metabolosomes.

## Results

**A system to investigate Pdu BMC biogenesis in S. Typhimurium LT2.** The genes encoding the proteins of the Pdu BMCs that mediate 1,2-PD degradation are located in the pdu operon (pduA-X) in the S. Typhimurium chromosome (Fig. 1a, b). Transcription of the pdu operon occurs in the presence of 1,2-PD during anaerobic growth and is activated by Crp and Arc[54,55]. To investigate the biogenesis pathway of Pdu BMCs in S. Typhimurium LT2, we designed a system based on the pBAD/Myc-His vector with the arabinose-inducible ParaBAD promoter[56,57] and labeled individual Pdu proteins with mCherry and super-folder GFP (sfGFP) (Fig. 1c). We used the sfGFP tag to visualize the assembly of the PduA shell protein and the PduE cargo protein. In the PduA-sfGFP background, we fused mCherry to other structural components (PduB/B′/J/K/M/N/T/U) and PduE individually. In the PduE-sfGFP background, we fused mCherry to the structural proteins (PduB/B′/J/K/M/N/T/U) and other cargo components (PduG/L/O/P/Q/S/V) individually (Supplementary Tables 1, 2).

To investigate the effect of increasing levels of protein expression, we detected the fluorescence of PduA-sfGFP at various L-arabinose concentrations in microcompartment-inducing media (MIM) (Supplementary Fig. 1). In the absence of a low concentration (0.00002%, g mL$^{-1}$) of L-arabinose, PduA-sfGFP appeared as discrete patches in cells, the typical subcellular distribution of Pdu BMCs as reported previously[24,58]. This expression of PduA-sfGFP probably resulted from the low-level transcription of the ParaBAD promoter in the absence of arabinose[59]. Higher concentrations of L-arabinose caused the formation of large puncta, showing that high levels of expression of Pdu proteins could cause aggregation or interference with the assembly of Pdu BMCs. Accordingly, the fluorescently-tagged Pdu proteins were expressed in S. Typhimurium LT2 without the addition of L-arabinose in this study.

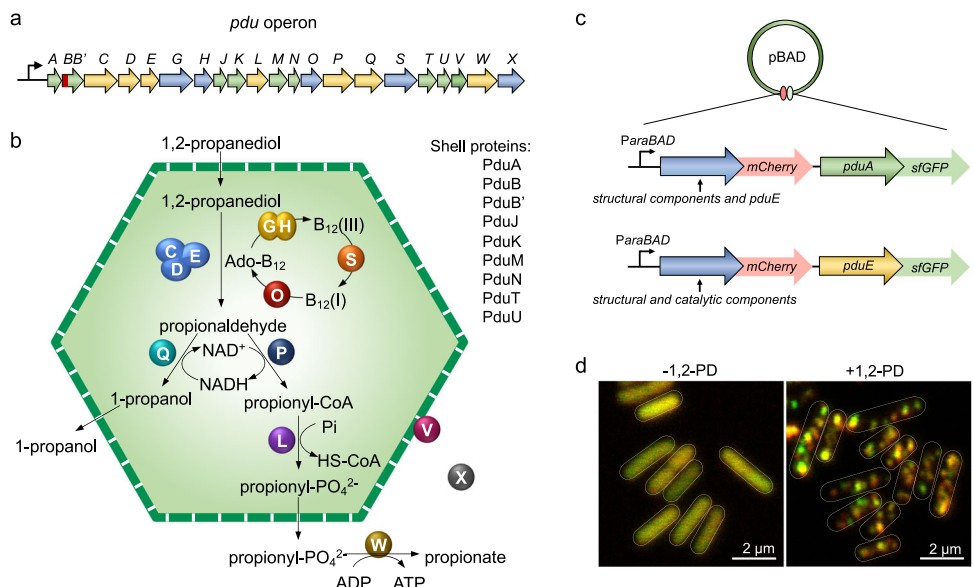

**Fig. 1 A system to investigate Pdu BMC biogenesis. a** The chromosomal *pdu* operon of *S.* Typhimurium LT2 encodes structural genes (*pduABB'JKMNTU*), catalytic genes (*pduCDELPQW* for 1,2-propanediol degradation, and the *pduGHOSX* B$_{12}$ recycling genes), and *pduV* (proposed to connect with filament-associated BMC movement[39]). PduM is listed as a structural protein according to the previous studies[28]. **b** The Pdu BMC shell encapsulates key enzymes that are required for 1,2-propanediol metabolism. **c** All Pdu proteins are visible by dual-labeling with fluorescence proteins (mCherry and sfGFP). Two series of vectors were constructed. First, PduA (a major shell protein) was labeled with sfGFP, and other structural components or PduE (a major enzymatic component) were labeled with mCherry. Second, PduE was labeled with sfGFP, and the structural components or other catalytic components were labeled with mCherry. **d** The presence of 1,2-propanediol (1,2-PD) is required to visualize Pdu BMCs. The fluorescence images show WT *S.* Typhimurium LT2 carrying pBAD-PduE-mCherry/PduA-sfGFP grown in microcompartment-inducing media (MIM), in the absence (−1,2-PD) or presence (+1,2-PD) of 1,2-propanediol.

Growth of *S.* Typhimurium LT2 in the absence of 1,2-PD does not stimulate Pdu BMC formation[16,24]. In our experimental model, growth in minimal media that lacked 1,2-PD (MIM-1,2-PD) allowed the visualization of expressed Pdu proteins diffused throughout the cytosol, confirming no formation of Pdu BMCs (Fig. 1d and Supplementary Fig. 2). Growth in the presence of 1,2-PD induced the expression of endogenous Pdu proteins and the formation of Pdu BMCs with a typical clustered distribution, as indicated by in vivo colocalization of PduE-mCherry and PduA-sfGFP fluorescence (Fig. 1d and Supplementary Fig. 3). This system and the generated *S.* Typhimurium LT2 mutant strains provided an ideal platform to explore the assembly and roles of individual Pdu proteins during Pdu BMC biogenesis.

**Roles of shell proteins during Pdu BMC biogenesis.** To characterize the roles of shell proteins in Pdu BMC assembly, we systematically generated a series of mutants that lacked individual shell proteins (Δ*pduA*, Δ*pduB*[1−37], Δ*pduB'*, Δ*pduJ*, Δ*pduK*, Δ*pduM*, and Δ*pduN*, Fig. 2a and Supplementary Fig. 4). pBAD-PduE-mCherry/PduA-sfGFP was then transformed into the deletion mutants, and Pdu BMCs were induced by growth in the presence of 1,2-PD, to investigate the assembly of cargos and shell proteins during Pdu BMC formation. For the Δ*pduA* strain, we specifically used PduE-mCherry/PduJ-sfGFP. The correlation between the assembly of shell proteins and cargos was evaluated by live-cell confocal imaging and colocalization analysis of sfGFP and mCherry fluorescence (Fig. 2b, c and Supplementary Fig. 5).

As a control, we investigated PduE-mCherry and PduA-sfGFP in the WT background and found that the individual proteins were colocalized into discrete patches (Fig. 2b, c). Correlation between PduE-mCherry and PduA-sfGFP (PduJ-sfGFP for Δ*pduA*) were also observed in the Δ*pduA*, Δ*pduB'* (the start codon of PduB' was replaced by GCT), Δ*pduJ*, Δ*pduK*, and Δ*pduN* strains (Fig. 2b, c). The Pdu BMC assemblies in the

Δ*pduA* and Δ*pduB'* cells possessed a similar distribution to that seen in the WT background. The absence of PduJ, the most abundant shell protein of the Pdu BMC[24], resulted in the formation of polar aggregates and occasionally elongated Pdu BMC structures (Fig. 2b, arrows), confirming that PduJ is required for both shell encapsulation and Pdu BMC formation[27]. Complementation experiments by expressing PduJ in the Δ*pduJ* strain showed similar growth of the WT and mutant (Supplementary Fig. 6); while elongated Pdu BMC structures were observed in the mutant cells, suggesting partially rescued assembly of Pdu BMCs, which was likely due to the non-physiological expression level of PduJ. In the absence of PduK, Pdu BMC assemblies were mostly located at the cell poles, implying a role for PduK in the spatial localization of Pdu BMCs (see detailed analysis below). In Δ*pduN*, the Pdu BMC assemblies possessed elongated structures predominantly, revealing that the pentameric protein PduN shaped the Pdu BMC polyhedral architecture in the same way as CcmL in the β-carboxysome[60]. In the Δ*pduB*[1−37] strain, in contrast, PduE assembled to form a large aggregate at one of the cell poles, whereas PduA formed several assemblies in the cytosol (Fig. 2b, c), explicitly revealing that the assembly of shell and cargo proteins were spatially separated. In the Δ*pduM* cells, PduE also formed a polar aggregate but features a relatively higher level of colocalization with PduA compared with that in Δ*pduB*[1−37] (Fig. 2b, c). These results indicate that PduB[1−37] and PduM play important roles in correlating assembly between the shell and cargo proteins of Pdu BMCs.

The Pdu BMC shell acts as a selectively permeable barrier that controls the influx/efflux of substrates and cofactors[6,19,61]. To evaluate the effects of shell proteins on shell integrity and enzyme activity, we performed cell growth assays on the WT and gene-deletion mutants in the presence of 1,2-PD at a limiting level of vitamin B$_{12}$ (Fig. 2d, see Methods). The activity of PduCDE is dependent on the vitamin B$_{12}$ level, the absence or defect of the

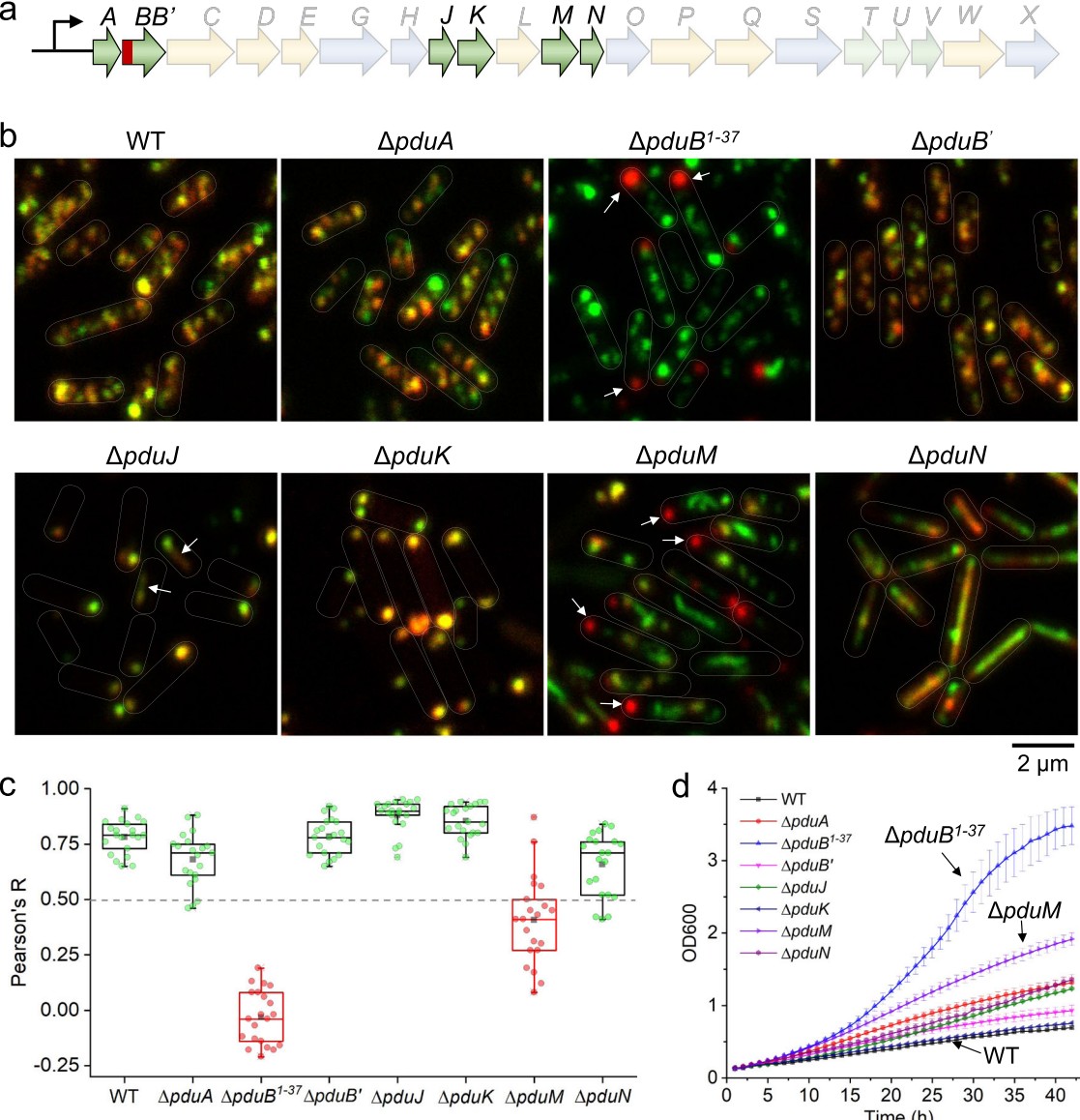

**Fig. 2 Role of *pdu* shell gene products in Pdu BMC biogenesis. a** Relevant shell-encoding genes of relevance to (**b**–**d**) are highlighted in the context of the *pdu* operon of *S.* Typhimurium LT2. **b** PduA-sfGFP (green, PduJ-sfGFP for Δ*pduA*) and PduE-mCherry (red) were visualized in WT and deletion mutants that lacked individual shell genes, following growth in MIM + 1,2-PD. **c** Colocalization analysis of sfGFP and mCherry fluorescence in (**b**). A Pearson's R value close to 1 reflects reliable colocalization, while values near zero indicate uncorrelated fluorescence distributions. Shell (PduA) and cargo (PduE) were colocalized in all strains except the Δ*pduB*[1–37] and Δ*pduM* mutants. The Pearson's R values for all strains were: 0.78 ± 0.07 (WT); 0.68 ± 0.12 (Δ*pduA*); −0.04 ± 0.12 (Δ*pduB*); 0.78 ± 0.08 (Δ*pduB'*); 0.88 ± 0.07 (Δ*pduJ*); 0.86 ± 0.07 (Δ*pduK*); 0.42 ± 0.19 (Δ*pduM*); 0.66 ± 0.14 (Δ*pduN*). Data are represented as mean ± SD. Boxplot center lines correspond to the median value; upper and lower whiskers extend from the box to the largest or smallest value correspondingly, but no more than 1.5 times the interquartile range; mean values show as the square symbol. $n = 20$, $n$ represents the number of cells. **d** Growth curves of WT *S.* Typhimurium LT2 and Pdu shell gene-deletion mutants on 1,2-PD with limiting B[12] (20 nM) in NCE medium (containing 0.6% 1,2-PD; 0.3 mM each of leucine, isoleucine, threonine, and valine; 50 μM ferric citrate). Data are represented as mean ± SD. $n = 4$, $n$ represents the number of biologically independent experiments.

shell structure caused by deletion of shell proteins may elevate the availability of the large molecule vitamin B[12] and thus increase the PduCDE activity[10,27]; the level of PduCDE activity enhancement is limited by the vitamin B[12] level, and the production of the toxic propionaldehyde is likely retained at a moderate level to ensure the mutants survive. These could explain why some Pdu shell gene-deletion mutants grow more rapidly than the WT at a limiting level of vitamin B[12] as reported previously[27].

At a limiting level of vitamin B[12], the Δ*pduB*[1–37] and Δ*pduM* strains exhibited significantly enhanced growth compared with other deletion mutants and the WT strain, confirming that the

assembly of shell and cargo proteins in the Δ*pduB*[1–37] and Δ*pduM* strains had been separated, as we visualized by confocal microscopy (Fig. 2b, c). The Δ*pduA*, Δ*pduB'*, Δ*pduJ*, and Δ*pduN* cells grew more rapidly than the WT, suggesting that the resulting Pdu BMC shells had altered structures to control the access and availability of vitamin B[12]. The growth rate of the Δ*pduK* mutant is comparable to that of the WT, confirming that PduK is not essential for controlling shell integrity[27,62].

However, during growth in the media containing a saturating level of vitamin B[12], the Pdu BMC shell no longer acts as a physical barrier that separates vitamin B[12] from the PduCDE

protein complexes, and so the PduCDE enzyme activity is maximized, which leads to enhanced propionaldehyde production. Cell growth assays at a saturating level of vitamin $B_{12}$ revealed that the intact Pdu BMC shells of WT *Salmonella* effectively sequestered propionaldehyde, as reflected by a higher growth rate; whereas the $\Delta pduB^{1-37}$ and $\Delta pduM$ mutants, in which cargo enzymes and shell proteins assemble separately, exhibited significant growth arrest between 18 and 32 h (Supplementary Fig. 7).

**Independent assembly of shell and cargo proteins and PduB$^{1-37}$-mediated binding between the shell and the enzymatic core.** Our results revealed that in the absence of the PduB N-terminus (PduB$^{1-37}$), the shell protein PduA and the cargo PduE were spatially disassociated and each of them self-assembled to form one or several aggregates (Fig. 2). To understand the composition of these protein assemblies, we studied the localization of individual Pdu proteins in the $\Delta pduB^{1-37}$ strain. Catalytic cargo components were tagged individually with mCherry in the PduE-sfGFP/$\Delta pduB^{1-37}$ background (Fig. 3a). When Pdu BMC expression was not stimulated (in the absence of 1,2-PD), the cargo proteins PduE, PduG, PduL, and PduQ were diffusely distributed throughout the cytosol (Fig. 3b, c). In contrast, both PduO and PduP formed aggregates at a single pole of the cell, accompanied by a low level of cytosolic distribution (Fig. 3b, c). In the presence of Pdu protein assemblies (following 1,2-PD induction), PduG, PduL, PduO, PduP, and PduQ colocalized with PduE at one pole of the $\Delta pduB^{1-37}$ cell (Fig. 3b, c and Supplementary Fig. 5), suggesting that the self-association of these Pdu cargo enzymes is required to form the enzymatic core located at the cell pole. Moreover, we determined the subcellular localization of multiple enzyme complexes, PduCDE and PduGH, in the absence of Pdu BMC structures formed in cells (Fig. 3d). The PduCDE and PduGH complexes were expressed from the pBAD plasmid, with PduE and PduH subunits tagged with sfGFP, respectively. PduCDE formed an aggregate close to the cell pole, whereas PduGH had a dispersed distribution in the cytosol, similar to PduG alone (Fig. 3b).

The aggregation of PduCDE-sfGFP and PduP-mCherry was likely mediated by the native encapsulation peptides located at the N-termini of the cargo proteins[63]. Further, we speculate that the aggregation of PduO reflects either the self-aggregation or insolubility of expressed PduO, or is a consequence of the fusion to the GFP protein. The difference between the fluorescence phenotypes of PduCDE-sfGFP (PduE tagged with sfGFP, Fig. 3d) and PduE-sfGFP (Fig. 3b, -1,2-PD) could reflect the fact that PduC and PduD were expressed from the pBAD vector instead of being endogenously expressed in the absence of 1,2-PD.

Collectively, our results show that intrinsic protein self-assembly is pivotal to the formation of the multi-subunit Pdu cargo complexes. We found that the fluorescence signal of the minor components PduS and PduV in the $\Delta pduB^{1-37}$ strain was weak (Supplementary Fig. 8), similar to the fluorescence observed in the WT background (Supplementary Fig. 3), thereby reflecting the low expression levels and partial distribution of these proteins in the cytosol. Furthermore, PduS and PduV did not colocalize with PduE (Supplementary Fig. 8), consistent with direct interactions between the PduS and PduV proteins and the shell[39,64].

To probe the shell assembling structures, individual shell components were labeled with mCherry in both the PduE-sfGFP/$\Delta pduB^{1-37}$ and PduA-sfGFP /$\Delta pduB^{1-37}$ background (Fig. 3e, f). Following the formation of Pdu BMCs, PduB′, PduJ, and PduK showed a patchy distribution; they colocalized with the shell protein PduA, but had no spatial correlation with the enzymatic

core (indicated by PduE-sfGFP fluorescence, Fig. 3f, g). The minor shell proteins PduN, PduT, and PduU[24] had a stronger propensity to colocalize with PduA than PduE (Supplementary Fig. 8). These results demonstrate that the shell proteins can self-associate to form multi-complex structures in the $\Delta pduB^{1-37}$ strain, which resemble the empty shell-like structures observed previously[43]. We corroborated the fluorescence data with thin-section EM, which identified a polar aggregate (likely the core structure) and several shell-like particles in the $\Delta pduB^{1-37}$ cell (Fig. 3h).

We detected the spatial separation of the assembly of shell proteins and cargos in the $\Delta pduB^{1-37}$ cells (Fig. 3); on the contrary, the absence of PduB′ had no major effects on the shell-cargo association (Supplementary Fig. 9). These findings pinpointed the importance of the PduB N-terminus itself in the physical binding between shell and cargo proteins. Complementation experiments by expressing PduB in the $\Delta pduB^{1-37}$ strain showed the recovery of Pdu BMC formation, as evidenced by confocal imaging, thin-section EM, and growth assays (Supplementary Fig. 10). As the $pduB$ gene is located at the front (5′ end) of the $pdu$ operon (Fig. 1a), the results could rule out the possibility of polar effects on the expression of downstream $pdu$ genes caused by our gene-deletion method, consistent with previous finding[65]. The formation of an enzymatic core without encapsulation of the shell explained the increased growth rate of the $\Delta pduB^{1-37}$ strain in a $B_{12}$-limiting environment, compared to that of the WT strain (Fig. 2d).

Taken together, our data demonstrate that the Pdu shell and cargo proteins have the intrinsic tendency to self-assemble resulting in the formation of independent shell-like structures and the enzymatic core. Moreover, PduB$^{1-37}$ serves as a linker peptide that acts as a bridge between the enzymatic core and shell assemblies, playing an essential role in the biogenesis of Pdu BMCs. Comparative genomic analysis revealed that PduB$^{1-37}$ is conserved throughout the *Klebsiella*, *Escherichia*, *Citrobacter*, and *Salmonella* genera that carry the $pdu$ operon (Supplementary Fig. 11). In these four genera, the PduB$^{1-37}$ linker protein was conserved as highly as the other structural or essential Pdu proteins (PduA/B/D/J/N/U). In contrast, the PduX protein, which has an unknown function, was not highly conserved[38]. Because the predicted structures of PduB$^{1-37}$ from different genera were very similar (Supplementary Fig. 11), our findings suggest that PduB$^{1-37}$ may play a ubiquitous role in the assembly of the Pdu BMC in different bacterial species.

**PduM plays a role in the binding between shell and enzyme core.** Our results showed that when the Pdu BMC shell-cargo association is impeded by the absence of PduB$^{1-37}$, PduM formed large puncta at the cell pole and colocalized with PduE but not with PduA (Fig. 3f, g). This suggests a strong interaction between PduM and cargo enzymes within the enzymatic core and no significant association between PduM and shell assemblies. To investigate the role of PduM in more detail, we studied the assembly of structural and catalytic Pdu proteins in the $\Delta pduM$ background (Fig. 4a, b). We found that the catalytic enzymes PduG, PduO, PduP, and PduQ colocalized with PduE, whereas the shell proteins PduB, PduB′, PduJ, and PduK assembled with PduA (Fig. 4b, c and Supplementary Fig. 12), recapitulating the results observed in the $\Delta pduB^{1-37}$ strain (Fig. 3). Moreover, confocal images of the $\Delta pduM$ strain expressing PduE-mCherry and PduA-sfGFP revealed a relatively low level of physical correlation between shell assemblies and cargos (Fig. 2b, c).

These results indicate that PduB$^{1-37}$ and PduM play similar roles in mediating the shell-cargo assembly. In addition, in the absence of PduM, both PduB and PduB′ colocalized with PduA in

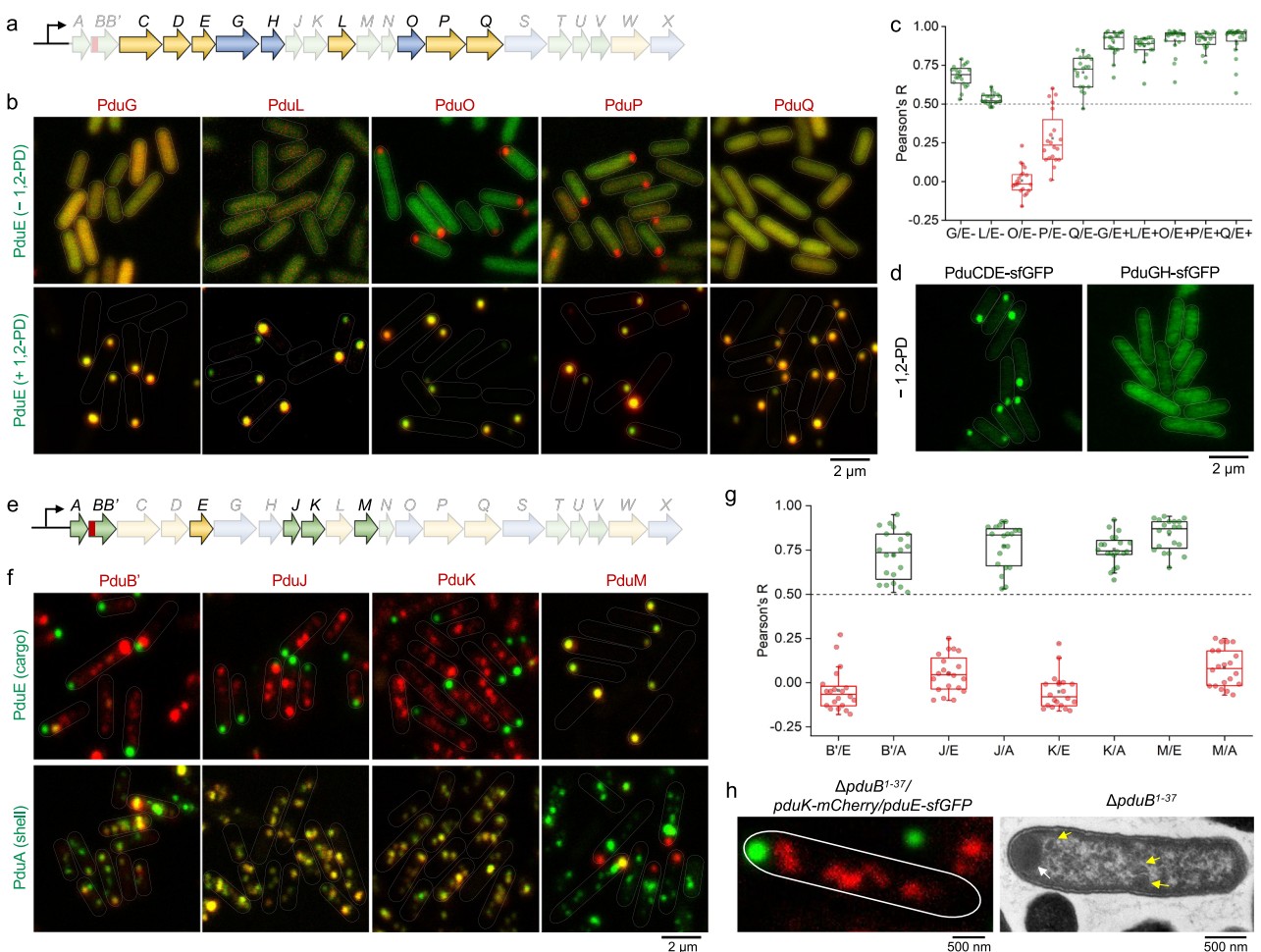

**Fig. 3 The PduB N-terminus links between the shell and the enzymatic core. a** Relevant cargo-encoding genes of relevance to Fig. 3b–d are highlighted in the context of the *pdu* operon of *S.* Typhimurium LT2. **b** PduE-sfGFP (green) and mCherry (red, labeling different catalytic components) were visualized in $\Delta pduB^{1-37}$ following growth in MIM in the absence and presence of 1,2-PD. **c** Colocalization analysis of sfGFP and mCherry fluorescence in (**b**). Note that in panels **c**, **g**, the first capital letter of the name refers to the Pdu protein tagged with mCherry and the second capital letter refers to the Pdu protein tagged with sfGFP. "+" and "−" represent the presence and absence of 1,2-PD, respectively. Pearson's R values for all the strains are: 0.68 ± 0.07 (G/E-); 0.53 ± 0.04 (L/E-); 0.01 ± 0.09 (O/E-); 0.28 ± 0.17 (P/E-); 0.70 ± 0.10 (Q/E-); 0.90 ± 0.08 (G/E+); 0.87 ± 0.07 (L/E+); 0.91 ± 0.08 (O/E+); 0.91 ± 0.05 (P/E+); 0.91 ± 0.10 (Q/E+). **d** The localization PduCDE and PduGH were characterized following growth in MIM-1,2-PD media. **e** Relevant genes encoding shell and catalytic proteins in **f**, **g**. **f** PduE-sfGFP (green) or PduA-sfGFP (green) and mCherry (red, labeling different structural components) were visualized in $\Delta pduB^{1-37}$, following growth in MIM + 1,2-PD media. **g** Colocalization analysis of sfGFP and mCherry fluorescence in (**f**). The Pearson's R values for all the strains are: −0.05 ± 0.12 (B'/E); 0.72 ± 0.14 (B'/A); 0.05 ± 0.10 (J/E); 0.77 ± 0.12 (J/A); −0.05 ± 0.10 (K/E); 0.75 ± 0.08 (K/A); 0.84 ± 0.09 (M/E); 0.08 ± 0.10 (M/A). **h** Microscopic characterization of the $\Delta pduB^{1-37}$ strain following growth in MIM + 1,2-PD media, showing a polar enzymatic core and several empty shell structures (arrows). Note: In **c**, **g**, data are represented as mean ± SD. $n = 20$, $n$ represents the number of cells. Boxplot center lines correspond to the median value, and upper and lower whiskers extend from the box to the largest or smallest value correspondingly, but no more than 1.5 times the interquartile range; mean values show as the square symbol.

the resulting shell assemblies, suggesting that PduM sits between PduB$^{1-37}$ and cargo enzymes to promote the shell-cargo binding. However, the function of PduM in the shell-cargo assembly appears less significant, as Pearson's R was higher in the absence of PduM ($\Delta pduM$) than that in the absence of PduB N-terminus ($\Delta pduB^{1-37}$) (Fig. 2c). This would be consistent with other associations occurring between PduB$^{1-37}$ and cargos, such as the binding of PduB$^{1-37}$ to cargo proteins via specific encapsulation peptides at the N-termini of enzymes localized within the Pdu BMC[40–42].

In addition to the shell-like structures formed by PduA, PduB, PduB', PduJ, and PduK in the $\Delta pduM$ mutant, some elongated or enlarged shell assemblies were also seen (Fig. 4b, d). This observation suggests that PduM is required for the defined

polyhedral structure of the Pdu BMC, which fits with previous findings[28]. Expressing PduM in the $\Delta pduM$ strain partially rescued the assembly of Pdu BMCs, as evidenced by a higher colocalization and a slower growth rate compared with those of $\Delta pduM$ (Supplementary Fig. 10). Interestingly, the aberrant assembly structures were still observed in the complementation experiment (Supplementary Fig. 10), perhaps reflecting the high level of PduM expression. It has previously been shown that only a low amount of PduM is required to define the assembly of native Pdu BMCs[24,28]. We searched for orthologous PduM proteins amongst 61 bacterial genomes that contain the *pdu* operon, and found that PduM was conserved (>50% identity at the amino acid level) in all but one strain (Supplementary Fig. 11). The high similarity of the predicted PduM structures

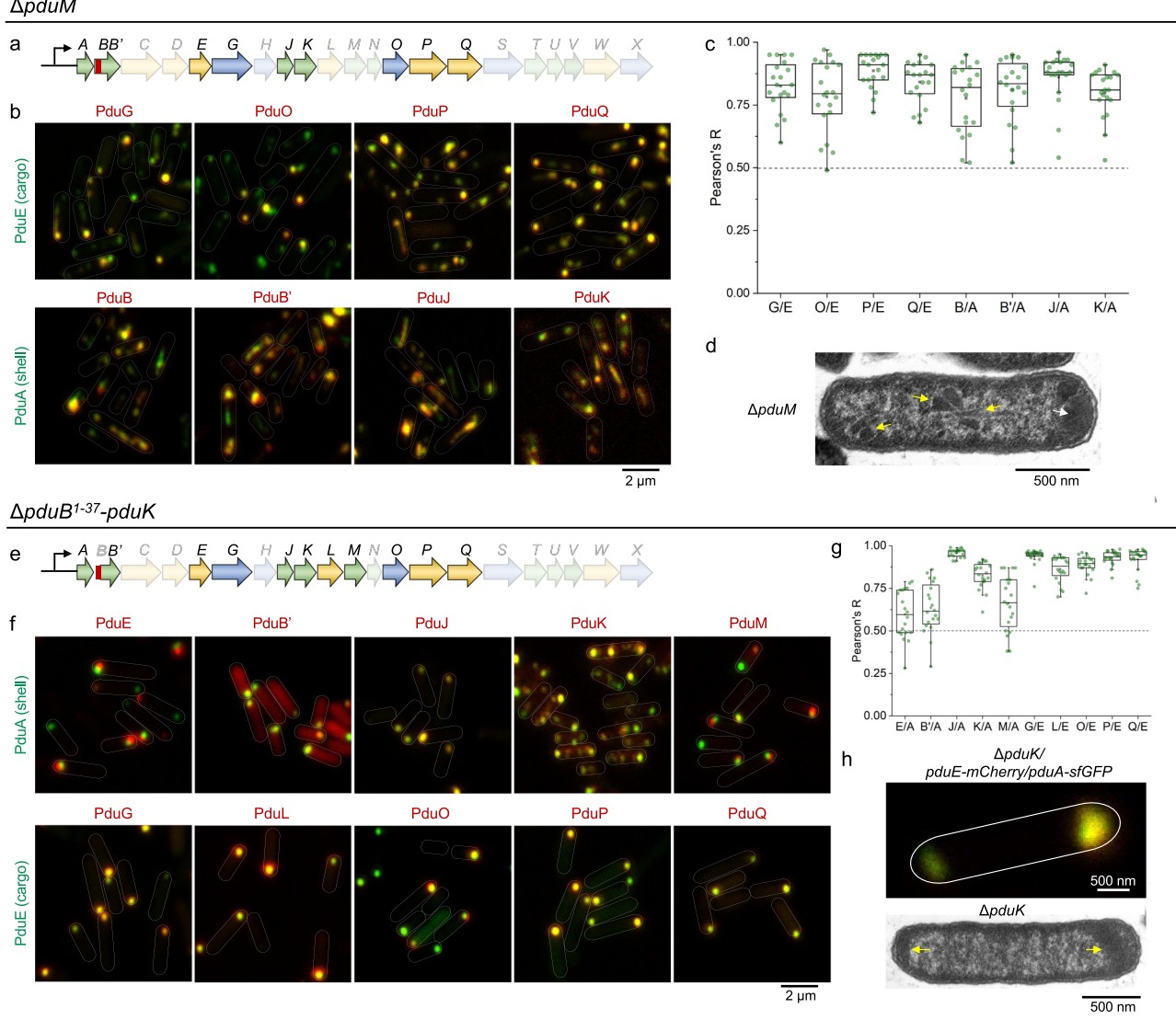

**Fig. 4 Roles of PduM and PduK in the biogenesis of Pdu BMCs. a** Relevant genes encoding shell and catalytic proteins of relevance to (**b**, **c**), highlighted in the context of the *pdu* operon of *S*. Typhimurium LT2. **b** PduE-sfGFP (green) and mCherry-fused catalytic components (red), and PduA-sfGFP (green) and mCherry-fused structural components (red) were visualized in Δ*pduM* grown in MIM + 1,2-PD media. **c** Colocalization analysis of sfGFP and mCherry fluorescence in (**b**). The graph is labeled as described in the legend to Fig. 3. The Pearson's R values for all the strains are: 0.83 ± 0.10 (G/E); 0.78 ± 0.14 (O/E); 0.89 ± 0.07 (P/E); 0.84 ± 0.08 (Q/E); 0.78 ± 0.13 (B/A); 0.81 ± 0.12 (B'/A); 0.86 ± 0.10 (J/A); 0.80 ± 0.09 (K/A). **d** EM of the Δ*pduM* strain, showing a polar enzymatic core (white arrow) and various shapes of the shell structure (yellow arrows). **e** Related shell and catalytic genes in **f**, **g** were highlighted in the *pdu* operon. **f** PduA-sfGFP (green) and mCherry (red, labeling PduE and different structural components), and PduE-sfGFP (green) and mCherry (red, labeling different catalytic components), were visualized in Δ*pduB*^1–37-Δ*pduK* strain growth in MIM + 1,2-PD media. **g** Colocalization analysis of sfGFP and mCherry fluorescence in (**f**). The Pearson's R values for all the strains are: 0.60 ± 0.14 (E/A); 0.63 ± 0.15 (B'/A); 0.96 ± 0.02 (J/A); 0.83 ± 0.08 (K/A); 0.66 ± 0.16 (M/A); 0.94 ± 0.05 (G/E); 0.86 ± 0.07 (L/E); 0.89 ± 0.05 (O/E); 0.93 ± 0.04 (P/E); 0.92 ± 0.07 (Q/E). **h** The polar aggregates of Pdu proteins were visualized in confocal microscopy (top) and EM (bottom, yellow arrows) of the Δ*pduK* strain. Note: In (**c**, **g**), data are represented as mean ± SD. *n* = 20, *n* represents the number of cells. Boxplot center lines correspond to the median value, and upper and lower whiskers extend from the box to the largest or smallest value correspondingly, but no more than 1.5 times the interquartile range; mean values show as the square symbol.

from these genera (Supplementary Fig. 11) suggests that PduM may play a critical role in Pdu BMC assembly.

**PduK is vital for the subcellular distribution of Pdu BMCs.** PduK is another minor shell protein within the Pdu BMC[24]. The N-terminal region of PduK has a high sequence similarity to the hexameric shell protein PduA, and the C-terminal extension of PduK has an unknown function (Supplementary Fig. 11). In the absence of PduK, Pdu BMCs were restricted to the cell poles (Fig. 2b)[27], and SDS-PAGE and EM results revealed that the Pdu

BMC structures are well-formed in the Δ*pduK* cells (Supplementary Fig. 12), suggesting that PduK is essential for the spatial localization of Pdu BMCs.

To explore the role of PduK in the biogenesis of Pdu BMCs, we determined the distribution of individual Pdu proteins in the Δ*pduB*^1–37/Δ*pduK* mutant (Fig. 4e). The major shell proteins PduB' and PduJ colocalized with PduA, forming assemblies at one cell pole (Fig. 4f, g and Supplementary Fig. 13). Apart from those colocalized with PduA, some PduB' proteins were diffusely distributed throughout the cytosol of the Δ*pduB*^1–37/Δ*pduK* cells,

which was also seen in the WT background (Supplementary Fig. 3). These findings are consistent with the comparison of Pdu protein abundance in cell extracts and isolated Pdu BMCs that indicated that a portion of PduBB' could not be assembled into Pdu BMC structures[24].

The localization of cargo enzymes in $\Delta pduB^{1-37}/\Delta pduK$ was comparable to that seen in $\Delta pduB^{1-37}$: PduG, PduL, PduO, PduP, and PduQ colocalized with PduE within a large aggregate located at one cell pole (Fig. 4f, g), indicating that PduK is not essential for the formation of the enzymatic core. However, unlike the $\Delta pduB^{1-37}$ cells that showed several patches of shell assemblies distributed in the cytosol, $\Delta pduK$ shell proteins were prone to assemble into large aggregates located at the cell poles (Fig. 4f, g). As depicted in PduA-sfGFP/PduE-mCherry/$\Delta pduK$ and PduA-sfGFP/PduM-mCherry/$\Delta pduK$, the shell and cargo aggregates were either located at the opposite poles of the cell or next to each other at the same pole but not exactly colocalized, suggesting that PduK plays a role in facilitating the relocation of in situ synthesized shell proteins for the generation of additional shell structures in the cytosol.

Consistent with the fluorescence results, EM showed polar aggregations of Pdu proteins in the $\Delta pduK$ strain (Fig. 4h), in agreement with previous observations[27]. Complementation experiments showed that expressing PduK in the $\Delta pduK$ strain rescued the distribution of Pdu BMCs, and the growth rate remained similar to WT (Supplementary Fig. 10). Expression of PduK-mCherry in $\Delta pduB^{1-37}/\Delta pduK$ also recovered the aberrant distribution of Pdu BMC shells (Fig. 4f).

Taken together, our results confirmed that the assembly of Pdu BMC shell proteins and cargos occur independently in vivo, and revealed that PduK is important for the assembly and subcellular distribution of both shell assemblies and entire Pdu BMCs. PduK orthologs were highly conserved and are predicted to have significant structural similarity between the bacterial genera that carry the *pdu* operon (Supplementary Fig. 11), suggesting a universal role of PduK in Pdu BMC assembly.

**A combination of two biogenesis pathways of the Pdu BMC**. To understand the precise nature of the Pdu BMC biogenesis, we studied the temporal order of the initial assembly of Pdu shell proteins and cargos, by applying time-lapse confocal imaging on cells that express either PduE-mCherry/PduA-sfGFP or PduJ-mCherry/PduE-sfGFP for cross-validation. After 1 h 1,2-PD induction to stimulate the expression of Pdu BMCs, fluorescence foci of shell proteins (indicated by PduA-sfGFP or PduJ-mCherry) and cargo proteins (indicated by PduE-mCherry or PduE-sfGFP) appeared at the cell poles from the cytosolic-distribution fluorescence background. In some cells, the fluorescent spot of shell proteins appeared earlier than that of cargo proteins, termed the "Shell first" event (Fig. 5a), whereas in some cells the fluorescent spot of enzymes formed first, termed the "Cargo first" event (Fig. 5a, b). Image analysis revealed that the "Shell first" events occurred as frequently as the "Cargo first" events (Fig. 5c, $n = 72$ vs 71 and $n = 74$ vs 75, respectively). Time-lapse fluorescence microscopy confirmed that the "Shell first" and "Cargo first" events co-occurred in different cells after 1 h 1,2-PD induction. The colocalization of shell and cargo fluorescence was then visualized after 1.5 h induction with 1,2-PD, indicating the subsequent import of cargo enzymes and encapsulation of shell structures, respectively, prior to the formation of the Pdu BMCs (Fig. 5d). We also observed the colocalization of shell and core assemblies occurring at 1 h, and speculate that a concomitant assembly or a fusion event of shell/cargo-independent assembly has occurred (Fig. 5d and Supplementary Table 4). Approximately 70.4% of the initial assembly

events occurred between the pole and the quarter positions along the longitudinal axis of the cell (Fig. 5e), reminiscent of the "birth" events of β-carboxysomes in cyanobacteria[48].

Overall, our data prove that the Pdu BMC undertakes both the Shell-first and Cargo-first assembly pathways, which is preceded by initial shell and cargo assembly that occurs independently and focuses at the cell poles.

**PduCDE triggers hierarchical assembly of the enzymatic core**. Our data showed that the cargo enzymes (PduCDE/PduGH/L/Q) have an inherent affinity to assemble to form the multi-complex enzymatic core (Fig. 3b, d), providing a means for enhancing catalytic efficiency of 1,2-PD degradation pathways of native Pdu BMCs. To investigate the mechanisms underlying the self-assembly of cargo enzymes, we characterized the subcellular locations of cargo proteins after removing the major enzyme PduCDE or minor enzymes PduO/PduP/PduQ/PduS in both the WT and $\Delta pduB^{1-37}$ background (Fig. 6a). Without PduCDE in the WT background with the addition of 1,2-PD, only PduE is distributed throughout the cytosol (Fig. 6b), reminiscent of previous findings suggesting that the PduD N-terminus is important for directing PduCDE into the Pdu BMC[41]. Both PduG and PduQ formed large puncta in addition to substantial cytosolic distribution, differing from the typical distribution of Pdu BMCs in WT. PduL is still assembled to construct several aggregates in most cells, given that the N-terminus of PduL serves as an encapsulation peptide[42]. Removal of both PduB[1-37] and PduCDE resulted in the cytosolic distribution of PduG and PduQ, differing from the polar aggregate in the $\Delta pduB^{1-37}$ strain (Fig. 3b). In contrast, the localization of PduL was not greatly affected (Fig. 6c).

On the contrary, the deletion of PduO/PduP/PduQ/PduS in the WT background did not affect the assembly of PduCDE, PduGH, and PduL within Pdu BMC structures (Fig. 6d). After removal of PduB[1-37] and PduO/PduP/PduQ/PduS, PduG, and PduL still formed a polar aggregate within the enzyme core (Fig. 6e), indicating that PduO/PduP/PduQ/PduS are not involved in the assembly of PduGH and PduL enzymes.

Together, our data shed light on the hierarchy of cargo assembly during the formation of the higher-ordered enzyme core and demonstrate that self-assembly of the major cargo enzyme complex PduCDE triggers the physical association of other cargo enzymes.

**Liquid-like association of internal enzymes within the Pdu BMC**. What is the organizational status of the enzymatic core made of multiple cargo complexes within the Pdu BMC? To tackle this question, we determined the in vivo diffusion dynamics of internal enzymes and shell proteins of Pdu BMCs, using fluorescence recovery after photobleaching (FRAP). We chose the elongated Pdu BMC structures in the $\Delta pduN$ strain (Supplementary Fig. 14), which are suitable for photobleaching and quantitative analysis[66]. Growth assays suggest that the resulting Pdu BMCs in $\Delta pduN$ are metabolically functional (Fig. 2d). Figure 6f, g show the representative FRAP image sequences of PduE-sfGFP (cargo) and PduA-sfGFP (shell) in the $\Delta pduN$ cells. The cargo enzymes exhibited a distinctly higher mobile fraction ($83 \pm 6\%$, $n = 20$), compared with shell proteins that showed a lower mobile fraction ($6 \pm 3\%$, $n = 20$) (Fig. 6f, g and Supplementary Table 4). The average diffusion coefficient of cargos within the Pdu BMCs of $\Delta pduN$ is $4.02 \pm 1.89 \times 10^{-4}$ $\mu m^2 s^{-1}$ ($n = 20$), more than 14-fold faster than that of shell proteins ($0.28 \pm 0.15 \times 10^{-4}$ $\mu m^2 s^{-1}$, $n = 20$).

In comparison, FRAP of the PduGH-sfGFP proteins that are dispersedly distributed in the cytoplasm (Fig. 3d) in the absence

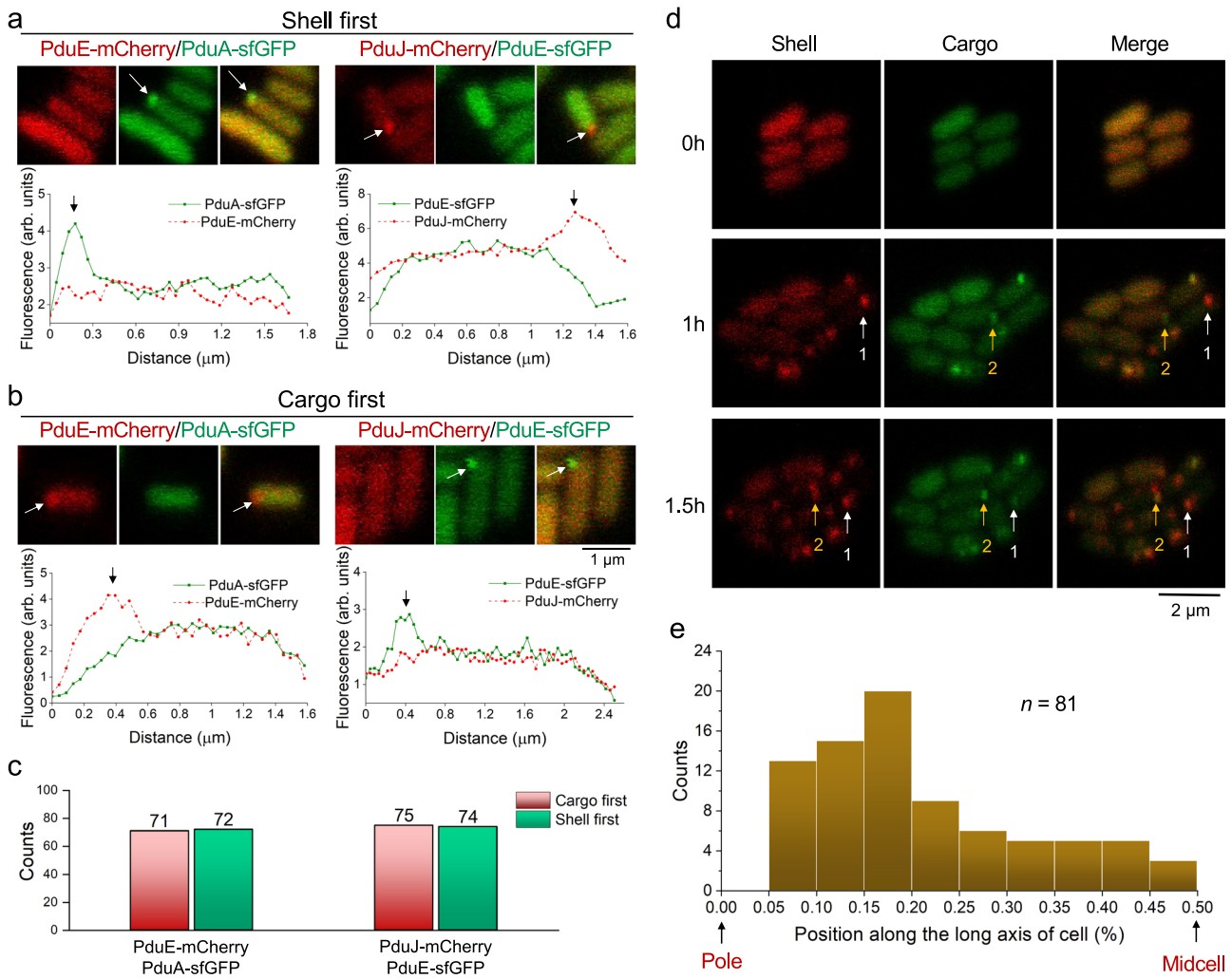

**Fig. 5 Pdu BMCs undergo a combination of the "Shell first" and "Cargo first" assembly pathways. a**, **b** Aggregation of the shell (PduA-sfGFP or PduJ-mCherry) and cargo (PduE-sfGFP or PduE-mCherry) in the WT strain following induction with 1,2-PD (addition of 0.6% 1,2-PD to MIM media at $t = 0$). Red: protein labeled with mCherry; green: protein labeled with sfGFP. **c** The distribution of the number of "Shell first" and "Cargo first" events in PduE-mCherry/PduA-sfGFP and PduJ-mCherry/ PduE-sfGFP. **d** Progression of shell and cargo formation in WT strain with pBAD-pduJ-mCherry/pduE-sfGFP. Red: PduJ-mCherry; green: PduE-sfGFP. "Shell first" (white arrow, number 1) and "Cargo first" (yellow arrow, number 2) events co-occurred 1 h after addition of 1,2-PD to MIM media. Shell and cargo fluorescence colocalized at 1.5 h after the addition of 1,2-PD (white and yellow arrows). **e** Histogram of the spatial distribution of the first Pdu BMC along the long axis of the cell. $n = 81$, $n$ represents the number of cells.

of Pdu BMC formation revealed that the PduGH-sfGFP fluorescence recovered rapidly after only ~3 s (Supplementary Fig. 15 and Supplementary Table 4), indicating that cargo enzymes exhibited highly dynamic mobility in the cytosolic solution than inside the Pdu BMC. While protein interactions and the internal arrangement of bacterial metabolosomes may differ between the canonical Pdu BMCs in the WT and the elongated structures of the Pdu BMCs in the $\Delta pduN$ strain, our results reveal the organizational dynamics of internal enzymes relative to the stable shell structure, suggesting that the cargo condensate within the Pdu BMC has a liquid-like property. A similar liquid-like state has been proposed to be the foundation of the assembly of α- and β-carboxysomes[47,67].

## Discussion

Pdu BMCs are self-assembling protein-based organelles (~202 MDa) composed of more than ten thousand protein polypeptides that sequester toxic intermediates to provide a key metabolic capability and reduce cytotoxicity[19,24,25]. The ability to

utilize 1,2-propanediol is required for the gastrointestinal pathogenesis of S. Typhimurium and other pathogens[20,68]. However, it has remained unclear how the large set of protein components of the Pdu BMC self-recognize and assemble so rapidly and efficiently to form such defined, functional architecture.

Previous attempts using genetic, biochemical, and microscopic analysis, and synthetic biology provided fragmented knowledge about the protein components essential for the assembly and structures of Pdu BMCs[27,31,39–41,43,69,70]. Here, we developed experimental and analytical tools to systematically study the roles and hierarchical assembly of Pdu BMC proteins in the native host. This study provides the first insight into the defined, step-wise biogenesis pathway of bacterial metabolosomes. First, the Pdu cargo and shell components are assembled independently, and then the Pdu metabolosomes are formed by a combination of the "Shell first" and "Cargo first" assembly pathways. The ordered consolidation of internal enzymes drives the formation of a liquid-like cargo aggregate to provide the framework for organelle assembly.

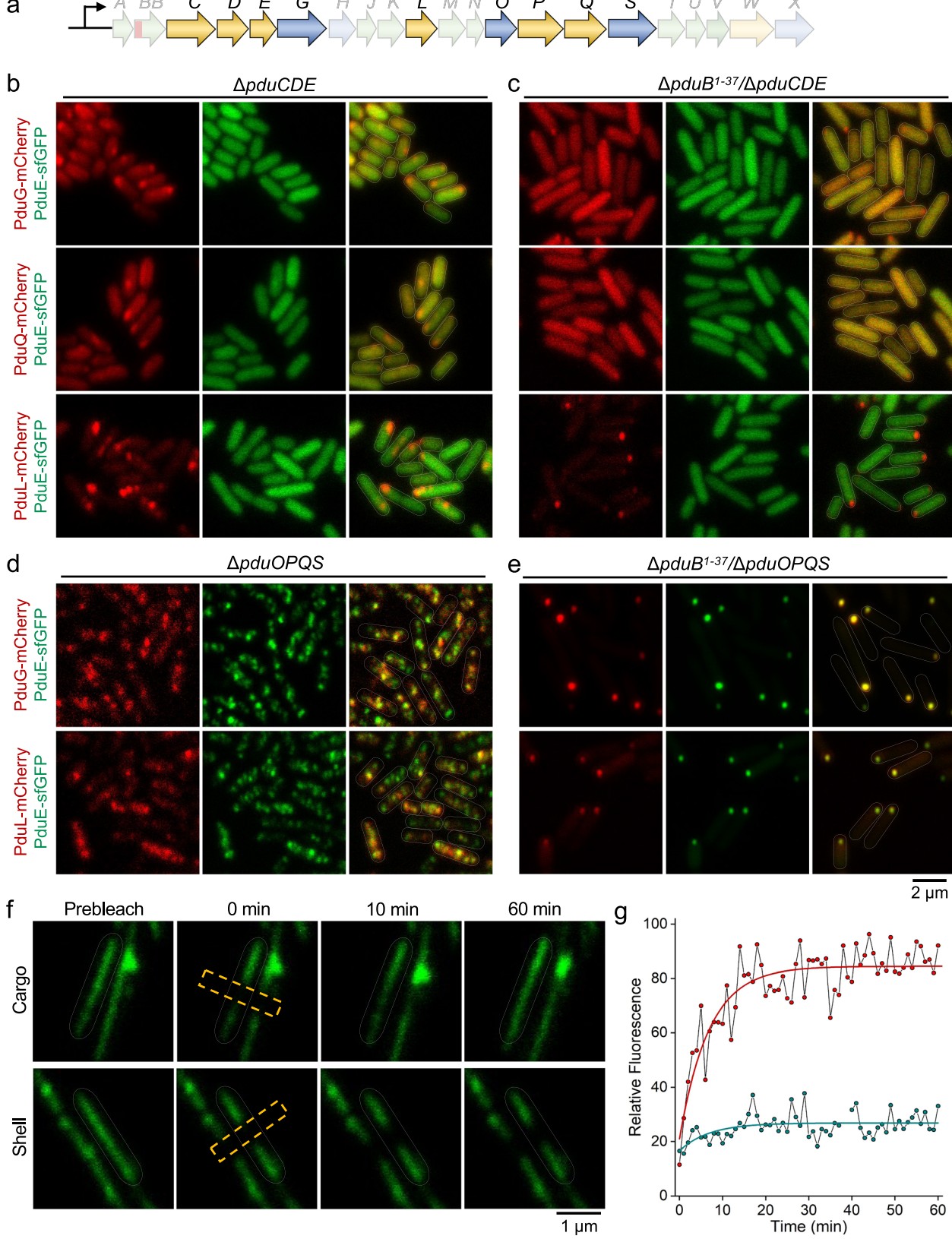

**A model for the hierarchical Pdu BMC biogenesis**. The ordered assembly of protein complexes has a general significance in many intracellular biological systems, including β-carboxysomes[44,71] and likely the glycyl-radical enzyme-associated microcompartments[72]. We propose a new model of Pdu BMC biogenesis based on our

findings that successive protein self-assembly and hierarchical association drive the formation of higher-order structural intermediates and entire Pdu BMCs (Fig. 7). The Pdu internal enzymes and shell proteins first self-associate independently to form cargo and shell aggregates. We found that shell proteins formed several

**Fig. 6 Interactions and liquid-like association of internal enzymes. a** Relevant genes encoding catalytic proteins of relevance to **b–e**, highlighted in the context of the *pdu* operon of *S*. Typhimurium LT2. **b, c** PduE-sfGFP (green) and mCherry (red, labeling different internal enzymes) were visualized in Δ*pduCDE* (**b**) and Δ*pduB*[1–37]/Δ*pduCDE* (**c**) grown in MIM + 1,2-PD media. **d, e** PduE-sfGFP (green) and mCherry (red, labeling different internal enzymes) were visualized in Δ*pduOPQS* (**d**) and Δ*pduB*[1–37]/Δ*pduOPQS* (**e**) grown in MIM + 1,2-PD media. **f** Representative fluorescence images before bleaching, immediately after bleaching of PduE-sfGFP (cargo) and PduA-sfGFP (shell), at various time lapses. The yellow rectangular boxes indicated the bleaching area. **g** Representative time course of fluorescence recovery of bleached regions of PduE-sfGFP (red) and PduA-sfGFP (cyan). The y-axis indicates fluorescence values relative to the fluorescence intensity of the selected region prior to bleaching. The recovery of sfGFP fluorescence is shown as circles and fitted to an exponential function.

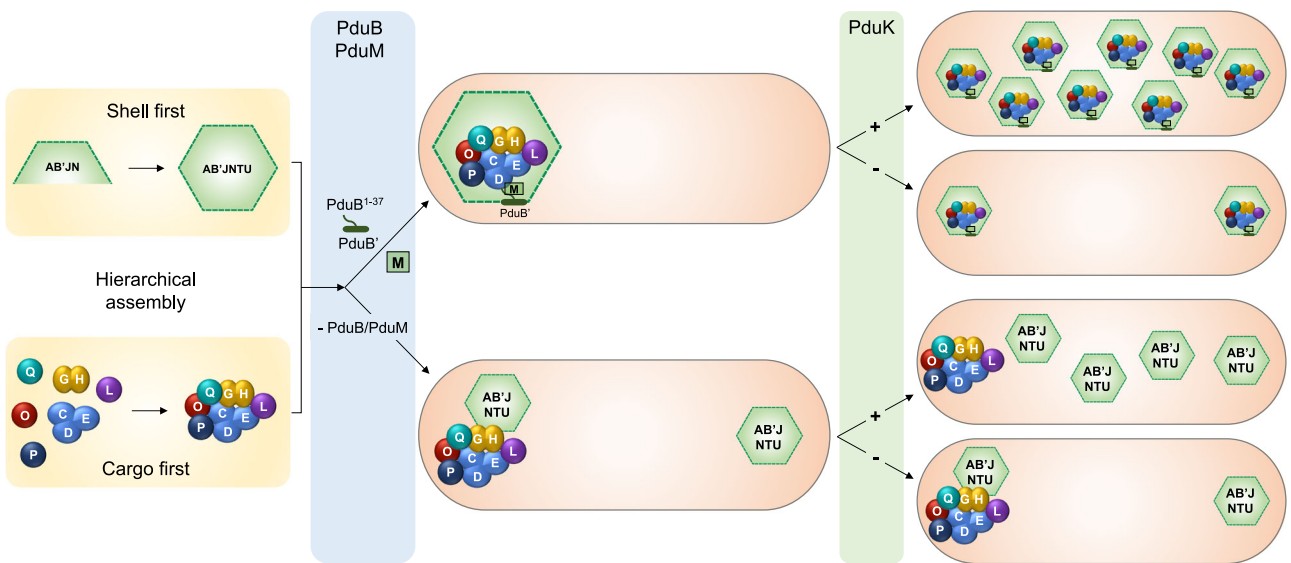

**Fig. 7 Model of the Pdu BMC assembly pathway of *S*. Typhimurium.** The Pdu internal enzymes and shell proteins independently self-associate to form cargo and shell aggregates, respectively, at the cell pole. PduB[1–37] binds the shell and cargo aggregates, mainly via PduM strongly associates with cargo enzymes. PduK is involved in the relocation of the assembled structural intermediates at the cell pole to other cellular positions, to permit the generation of additional Pdu BMCs.

assembly intermediates in the cytosol of *S*. Typhimurium LT2 cells in the absence of PduB[1-37], including empty shell structures (Fig. 3). It has been suggested that PduJ, PduN, PduA, and PduB could first assemble to form scaffolding structures to trigger the association of other shell proteins, as PduJ and PduN are essential for the formation of the shell and Pdu BMC structures[27], and PduA, PduB, and PduJ can form nanotube shell structures[31,73,74].

The enzymatic core of the Pdu BMC is clustered into six distinct types of enzyme complexes. Some internal enzymes, including PduP, PduD, and PduL, feature N-terminal extensions that serve as encapsulation peptides to ensure the recognition and loading of internal enzymes[40–42]. The initial assembly of the main PduCDE enzyme complex may create high-affinity binding sites for cargo enzymes through the N-termini of PduL and PduP, to promote the ultimate formation of the enzymatic core near the cell poles (Fig. 6).

The N-terminal short peptide of the shell protein PduB (PduB[1–37]) plays an essential role in binding the enzymatic core to the shell (Figs. 2, 3), as suggested earlier[43]. PduB[1–37] mainly interacts with the internal core through PduM; PduM in turn strongly binds with cargo enzymes, likely the N-terminus of PduD that serves as an encapsulation peptide. Both PduB[1–37] and PduM play multiple roles, being important for shell-interior interaction, shell encapsulation, and full assembly of intact Pdu BMC structures, reminiscent of the β-carboxysome CcmM and CcmN that serve as the essential linker proteins to bind the shell and Rubisco matrix during β-carboxysome assembly[44]. The fact that orthologs of PduB[1–37], PduK, and PduM were identified in *Klebsiella*, *Escherichia*, *Citrobacter*, and *Salmonella* suggests that

the general assembly principle of Pdu metabolosomes that we have identified may be pervasive among these genera (Supplementary Fig. 11), and might extend to other metabolosomes that have multiple interior enzymes possessing structurally-similar encapsulation peptides.

Although diverse BMCs share a common shell architecture, the assembly pathways of the distinct types of BMCs differ[75]. The biogenesis of β-carboxysomes adopts a "Cargo first" pathway, with the assistance of chaperones[44–46,48]. The assembled Rubisco enzymes first nucleate by interactions with CcmM to form a preorganized cargo matrix, which is then encapsulated by shell proteins to form the functional carboxysome. In contrast, the assembly of α-carboxysomes may follow the "Shell first" or "concomitant shell–core assembly" pathway, as partially-formed α-carboxysomes have been visualized in vivo[50–52,76], and empty α-carboxysome shells have been reconstituted in heterologous hosts[53]. In the present study, both pathways were visualized equally during live-cell confocal imaging (Fig. 5). These results provided the first evidence that both the "Shell-first" and "Cargo-first" assembly pathways are required in Pdu BMC biosynthesis. These findings are supported by the successful construction of empty Pdu BMC shells, and the fact that encapsulation peptides are involved in Pdu BMC assembly[39,41]. During the shell-first and cargo-first events that underpin the formation of mature Pdu BMCs, we hypothesize that shell or cargo proteins begin by forming partially assembled structures, which then dynamically associate with cargo enzymes or shell proteins, respectively. Such a mechanism would be supported by the partially-formed α-carboxysomes that have been observed previously[50].

**De novo biogenesis of Pdu metabolosomes occurs at the cell poles**. We found that generation of the Pdu BMC enzyme and shell assembly intermediates generally occurred at a cellular pole. Consistently, the procarboxysome structures formed during β-carboxysome biogenesis and degradation of inactive β-carboxysomes also occur in the polar regions of cyanobacterial cells[44,77]. Likewise, α-carboxysomes were observed exclusively at the cell poles of chemoautotrophs when lacking the capacity for functional positioning in the absence of the McdAB system[78]. These observations suggest that the cell pole represents a universal, spatially-confined region for BMC biogenesis in bacterial cells.

The intracellular positioning of BMCs is of physiological importance for cellular metabolism and can ensure equal segregation of BMCs during cell division[79,80]. It has been proposed that specific interactions between carboxysomes and the cytoskeletal protein ParA and McdAB mediate the partitioning of both α- and β-carboxysomes in cyanobacteria and proteobacteria[78,79,81,82]. Our results suggest that PduK is involved in the relocation of the assembled structural intermediates from the cell pole to other cellular positions to generate additional Pdu BMCs. The PduK component of the Pdu BMC could interact with other intracellular elements via its C-terminal flexible extension at the concave side facing outside of the shell, to control the spatial positioning and mobility of Pdu BMCs in bacterial cells.

**The internal enzyme aggregate within Pdu metabolosomes possesses a liquid-like nature**. Liquid–liquid phase separation (LLPS) has been increasingly recognized as a general mechanism that mediates the formation and organization of protein assemblies and membrane-free organelles[83–85]. LLPS has also been proposed to play important roles in determining the cytoplasmic behavior of bacterial cells, with broad impacts on the metabolism of Enterobacteriaceae[86]. The intriguing features of LLPS involve the local cytosolic accumulation of cargo molecules and the presence of intrinsically-disordered protein peptides. Cargo nucleation and condensation are mediated by specific multivalent weak interactions between the intrinsically-disordered protein peptides and structural proteins[85,87].

The Pdu metabolosome contains several short N-terminal protein extensions that function as encapsulation peptides for the recruitment of multiple cargo enzyme complexes and represent a model system for the assessment of the LLPS-mediated ordered assembly of protein organelles. We note that the secondary structures of these encapsulation peptides (an amphipathic α-helix) do not perfectly meet the definition of conventional proteins that undergo LLPS, which often contain large disordered regions of low-complexity, composed of repetitive and biased amino acids[63,70,88,89], and might represent a novel class of LLPS systems.

Our data demonstrate the hierarchical assembly of Pdu enzyme complexes and provide in vivo experimental evidence suggesting that the cargo enzymes form a dynamic protein condensate within the Pdu BMC. The organizational dynamic rate of Pdu cargo proteins is lower than that of free GFP in the Escherichia coli (E. coli) cytosol[90] and that of membrane complexes in thylakoids[91]. We speculate that Pdu metabolosomes leverage the flexible and dynamic internal organization to gain a balance between stable association and fluidity of cargo enzymes. Such a liquid-like nature would enhance enzyme activities of the catabolic pathways and would permit reorganization of internal proteins and metabolites between the multiple sequestered pathways to modulate metabolism in the fluctuating environments experienced by Salmonella bacteria during infection[92].

A similar liquid-like cargo condensate mediated by linker proteins has been proposed for $CO_2$ fixation in anabolic BMCs and algal pyrenoids. In α- and β-carboxysomes, Rubisco enzymes form dynamic interactions with the intrinsically-disordered CsoS2 N-terminus and CcmM35 respectively, to generate the liquid-like condensate matrix of Rubisco within the carboxysome[47,67]. Similarly, within the $CO_2$-fixing pyrenoid organelles of eukaryotic algae, the liquid-like state of Rubisco clusters is facilitated by the intrinsically-disordered protein EPYC1[93,94]. LLPS-facilitated cargo sorting mechanisms have also been identified recently in plant chloroplasts[95].

This contextualization of our latest findings with other prokaryotic and eukaryotic organelles leads us to propose that LLPS plays a universal role in the formation of metabolic organelles that span diverse architectures, biogenesis pathways, and functions.

## Methods

**Bacterial strains and growth conditions**. The bacterial strains used in this study derived from S. enterica subsp enterica serovar Typhimurium LT2[96,97]. The rich medium was LB-Lennox medium (10 g L$^{-1}$ tryptone (Appleton Woods, MN649), 5 g L$^{-1}$ yeast extract (Appleton Woods, DM832), 5 g L$^{-1}$ sodium chloride, pH 7.0), and the minimal medium was no-carbon-E (NCE) medium[98]. The microcompartment-inducing media (MIM) was NCE medium supplemented with 1 mM MgSO$_4$, 0.5% succinate, 50 μM Fe(III) citrate, and 0.6% 1,2-PD (if applicable)[28,43]. The MIM medium does not contain vitamin B$_{12}$, as succinate serves as the carbon source instead of 1,2-PD and vitamin B$_{12}$ is not required for S. Typhimurium LT2 growth in MIM. All medium components were from Sigma-Aldrich, except where specified.

Intracellular visualization of fluorescently-tagged Pdu BMC proteins was done following the growth of the relevant plasmid-carrying strains in MIM containing 0.6% 1,2-PD (MIM + 1,2-PD) under aerobic conditions. An overnight LB culture was inoculated 1:100 in 100 μL of MIM in a 2-mL Eppendorf tube in the absence of 1,2-PD (MIM-1,2-PD) shaken horizontally at 220 rpm overnight (Innova 2300 Platform shaker). Unless otherwise specified, 1 μL of this culture was sub-inoculated to 100 μL of MIM in a 2-mL Eppendorf tube, both in the absence and in the presence of 1,2-PD, shaken aerobically at 37 °C for 10 h until OD$_{600}$ reaching 1.0 to 1.2 (DS-11 Spectrophotometer/Fluorometer Series from DeNovix); for birth event detection, the sub-inoculated culture was shaken aerobically at 37 °C for 1 h.

Antibiotics were added to liquid media as required at the following final concentrations: ampicillin at 100 μg mL$^{-1}$, kanamycin at 50 μg mL$^{-1}$, gentamicin at 20 μg mL$^{-1}$ in ddH$_2$O, chloramphenicol at 25 μg mL$^{-1}$ in ethanol, and tetracycline at 25 μg mL$^{-1}$ in methanol.

**Construction of chromosomal mutations and fluorescence tagging vectors**. LT2-ΔpduB$^{1-37}$ and LT2-pduB′ were constructed by genome editing technique developed previously[99]. The PduB$^{1-37}$ truncation (LT2-ΔpduB$^{1-37}$), resulting in the expression of the N-terminus truncated PduB (namely PduB'), was achieved by deleting the 5′ terminal region of pduB, without modification of the pduB′ gene promoter, RBS and starting codon as confirmed by DNA sequencing. LT2-pduB′ was achieved by replacing the start codon for PduB′ with GCT (alanine). The pEMG and pSW-2 plasmids were used. First, DNA fragments (700–800 bp) flanking the chromosome regions of interest were PCR amplified and inserted into the pEMG suicide plasmid by Gibson assembly (NEBuilder HiFi DNA Assembly kit). The pEMG-derivative suicide plasmids were mobilized from E. coli S17-1 λpir to S. Typhimurium by conjugation. S. Typhimurium transconjugants clones that have integrated the suicide plasmid were selected on solid M9 minimal medium supplemented with 0.2% of glucose and 50 μg/mL of kanamycin. pSW-2, purified from LT2-WT strain, was then introduced into transconjugants by electroporation. Transformants were selected on LB agar medium supplemented with 20 μg/mL gentamicin and 1 mM of m-toluate. Colonies were screened for kanamycin resistance and sensitive clones were tested by specific PCR. pSW-2 was cured from the resulting strains by two passages in LB in the absence of gentamicin.

Other Salmonella single mutants were constructed by the gene disruption method established previously[100]. pKD4 and pKD3 plasmid were used as the template for PCR. DNA fragments with antibiotic resistance cassette were PCR amplified and transformed into S. Typhimurium harboring pSIM5-tet[101] for lambda red recombination. P22 transduction was performed to move the individual mutation into a clean genetic background[102]. Then the antibiotic resistance cassette was eliminated using the FLP recombinase expressing plasmid pCP20. Thermo-sensitive pCP20 was cured from the resulting strains by two passages in LB in the absence of ampicillin at 42 °C. Double mutants were constructed following the same protocol but using ΔpduB$^{1-37}$ as the background strain instead of WT.

pBAD/Myc-His was used as a backbone for the construction of visible vectors. First, this plasmid was digested by NcoI and HindIII. Then PCR-amplified DNA

fragments of individual *pdu* genes, mCherry, and sfGFP were cloned into the linear vector by Gibson assembly[103]. pXG10-SF containing a constitutive P$_{LtetO-1}$ promoter was employed as a backbone of expression vectors for complementation experiments[104]. The linear pXG10-SF was obtained by PCR cloning. The coding sequences of PduB (M38A), PduB′, PduJ, PduM, and PduK were cloned into linear pXG10-SF via PCR and Gibson assembly.

The strains and plasmids used in this study are listed in Supplementary Table 1. A complete list of primers used is in Supplementary Table 2. All mutants and vectors were verified by PCR and DNA sequencing of PCR-amplified genomic DNA or plasmid sequencing (Supplementary Fig. 4).

**Bacterial growth assays.** Overnight LB cultures were inoculated 1:1000 in 10 mL of LB supplemented with 0.6% 1,2-PD and shaken aerobically in 50 mL Falcon tubes for 6 h at 220 rpm. The cells were washed three times with a mixture of 0.6% 1,2-PD and 1 mM MgSO$_4$ in the NCE medium and then resuspended in the NCE medium (containing 0.6% 1,2-PD; 0.3 mM each of leucine, isoleucine, threonine, and valine; 50 μM ferric citrate; and 20 nM or 150 nm CN-B$_{12}$) to an OD$_{600}$ of 0.15. At 20 nM of CN-B$_{12}$ (a limiting level of vitamin B12), 200 μL of the culture at OD$_{600}$ = 0.15 was grown aerobically at 37 °C with intermittent shaking in a microplate reader (FLUOstar Omega, BMG LABTECH).

At 150 nM CN-B$_{12}$ (saturating level of vitamin B12), 250 μL of the culture at OD$_{600}$ = 0.15 was grown aerobically at 37 °C with intermittent rapid shaking, using the System Duetz technology platform (Growth Profiler 960, EnzyScreen). The OD$_{600}$ readings were taken hourly. At least three biological replicates of each growth curve were obtained.

**Pdu BMC purification.** The Pdu BMCs from both *S.* Typhimurium LT2-WT and LT2-Δ*pduK* were isolated by detergent treatment and differential centrifugation[28]. Briefly, 400 mL cells grown in MIM (OD$_{600}$ = 1.0–1.2) were harvested and washed with buffer A (50 mM Tris-HCl pH 8.0, 500 mM KCl, 12.5 mM MgCl$_2$, and 1.5% 1,2-PD), and lysed with the bacteria-specific reagent (BPER-II, ThermoFisher, 78260). Subsequently, Pdu BMCs were separated from cell debris by sequential centrifuge steps (12,000×*g* for 5 min to pellet cell debris, 20,000×*g* for 20 min to pellet Pdu BMCs). The isolated Pdu BMCs were washed with a mixture of buffer B (50 mM Tris-HCl pH 8.0, 50 mM KCl, 5 mM MgCl$_2$, 1% 1,2-PD) and BPER-II, and resuspended in buffer B containing protease inhibitor cocktail (Sigma-Aldrich). Finally, isolated Pdu BMCs were obtained by centrifugation at 12,000×*g* (3 × 1 min) to further remove cell debris.

**SDS-PAGE analysis.** Standard procedures for sodium dodecyl sulfate-polyacrylamide gel electrophoresis (SDS-PAGE) were employed. Proteins were separated by 15% polyacrylamide gels and stained with Coomassie Brilliant Blue G-250.

**Transmission electron microscopy.** The *S.* Typhimurium LT2-WT and mutant cells were characterized using thin-section EM[80]. *Salmonella* cells were pelleted by centrifugation (6000×*g*, 10 min) and processed for thin sections using a Pelco BioWave Pro laboratory microwave system. The cells are first fixed with 0.1 M sodium cacodylate buffer (pH 7.2) supplemented with 2.5% glutaraldehyde using two steps of 100 W for 1 min each (P1). Samples were then embedded in 4% agarose, followed by staining with 2% osmium tetroxide and 3% Potassium Ferrocyanide using three steps of 100 W for 20 s each (P2). The reduced osmium stain was then set using a solution of 1% Thiocarbohydrazide for 10 min. The second osmium stain was applied using P2 with 2% osmium tetroxide. The sample was made electron dense with 2% Uranyl Acetate incubated at 4 °C overnight. Dehydration was operated with a series of increasing alcohol concentrations (30 to 100%) before cells embedding in medium resin. Thin sections of 70 nm were cut with a diamond knife. The structures of purified Pdu MCPs were characterized using negative staining EM as described previously [45,105–107]. The isolated Pdu BMCs were stained with 3% uranyl acetate. Images were recorded on an FEI 120Kv Tecnai G2 Spirit BioTWIN transmission electron microscope equipped with a Gatan Rio 16 camera and DigitalMicrograph software.

**Confocal microscopy.** The cells were imaged using a Zeiss LSM780 confocal microscopy with a 63× oil-immersion objective excitation at 488 and/or 561 nm. GFP and mCherry fluorescence were detected at 500–520 nm and 660–700 nm, respectively. Images were captured at 512 × 512 pixels at 16 bits by Zeiss Zen 2010 software. The pinhole was set to ~1 μm to make sure that all fluorescence signals of the *Salmonella* cells were recorded. The temperature was controlled at 37 °C during the whole imaging process. All images were captured from at least three different cultures. Colocalization analysis was performed by using the Coloc2 plugin in ImageJ to generate Pearson's correlation coefficient R and scatterplot[108]. The values of Pearson's R close to 1 reflect reliable colocalization, whereas the values near zero indicate distributions of fluorescence uncorrelated with one other[109]. Scatterplots are generated by plotting the intensity value of each pixel of mCherry along the x-axis and the intensity value of the same pixel location of sfGFP on the y-axis using Coloc2 plugins in ImageJ. The scatterplots describe the relationship between the fluorescent signals. If the dots on the diagram appear as a cloud clustered on a line indicates a strong colocalization, and Pearson's R is close

to 1 in this case. If the scattered distributions of the pixels close to both axes indicate a mutual exclusion, and Pearson's R is near zero. Fluorescence intensity profiles were generated using ImageJ software.

To monitor the progression of shell and cargo formation of Pdu BMCs, overnight MIM (−1,2-PD) culture was diluted 1:100 in MIM (+1,2-PD) preincubated at 37 °C and 10 μL of this culture was dropped onto MIM agar plate (+1,2-PD) and left to dry at 37 °C. The MIM agar with cell patches was cut out and attached to a 0.17 mm glass coverslip, followed by sandwiched within a 35 mm glass-bottom dish. The temperature of the microscope system was set to 37 °C for cell growth. The images were taken at 0 h when the cells started to grow in the presence of 1,2-PD, at 1 h when the shell and cargo started to assemble, and at 1.5 h for further tracking.

**Fluorescence recovery after photobleaching.** Δ*pduN*/pBAD-*pduE-sfGFP* and Δ*pduN*/pBAD-*pduA-sfGFP* cells in the stationary phase were diluted to 0.2 to 0.4 OD$_{600 nm}$, and 2 μL was spotted onto a 2% low-melting agarose ampicillin plate that was preincubated at 37 °C. After cells were dried, the pad was converted onto a glass-bottom dish (35 mm), which was covered by a cover glass. Then cells were imaged with a Zeiss LSM780 confocal microscopy. 100% laser power confocal laser was used to bleach a line across the center of the cell. Images were taken every 1 min for 60 min to ensure the stationary of fluorescence intensity. FRAP data were analyzed following the previous procedure[91,110]. The fluorescent profiles within a cell were obtained by ImageJ software and were normalized to the same total fluorescence to compare fluorescence distributions before and after bleaching. The mobile proportion (M) was given by Eq. (1):

$$M = \frac{(\text{Final fluorescence}) - (\text{Postbleach fluorescence})}{(\text{Scaled prebleach fluorescence}) - (\text{Postbleach fluorescence})} \quad (1)$$

Fluorescence profiles along the long axis of the cell were extracted and normalized to the same total fluorescence against that of the prebleached cell. This normalization aimed to minimize the fluctuations caused by the microscopy and the decrease of fluorescence intensity caused by laser scanning. Then we subtracted the scaled pre-bleach profile from the post-bleach profile to give a different profile, which allowed us to know the final fluorescence intensity. The different profile was plotted versus cell length to find the minimum value (usually the center of the bleached area in the x-position). The fluorescence profile values at the center of the bleached area were plotted against time to give the experimental fluorescence recovery curve. The single exponential function is given by Eq. (2):

$$f(t) = A(1 - e^{-\tau t}) \quad (2)$$

Half time is given by Eq. (3):

$$\tau_{1/2} = \frac{ln0.5}{-\tau} \quad (3)$$

Based on the post-bleaching and final fluorescence profiles, it is possible to simulate the evolution of the fluorescence profile. This was done by running the experimental different profile in a homemade and iterative computer routine on SigmaPlot14 software, assuming an arbitrary diffusion coefficient for random diffusion[91,111]. The routine can predict how the fluorescence profile will evolve in small time increments, based on the basic diffusion Eq. (4):

$$\frac{dC}{dt} = D\frac{\delta^2 C}{\delta X^2} \quad (4)$$

The routine estimates the value of $\delta^2 C/\delta X^2$ at each point by calculating the difference between points in the profile and its two neighbors. This value was then multiplied by a factor related to the diffusion coefficient and was used to predict the increment in C at that point. Where the mobile fraction is less than 1, the routine calculates the "mobile profile" and predicts how it will evolve. Then a set of predicted fluorescence profiles were generated to estimate the diffusion coefficient. Once the simulated fluorescence curve was overlapped with the experimentally observed fluorescence recovery curve, the diffusion coefficient of the simulation was recorded.

**Bioinformatic analysis and structural prediction.** To identify the bacterial genomes that harbor the Pdu operon, the protein sequence of PduA was queried against the complete genomes (*n* = 23,521) downloaded from the Refseq bacterial database (according to https://www.ncbi.nlm.nih.gov/genome/doc/ftpfaq/) using tBLASTn v2.5.0+ [112]. Four bacterial genera were found to carry PduA gene: *Citrobacter*, *Escherichia*, *Salmonella*, and *Klebsiella*. From the four genera, 61 strains were selected to represent different species or sub-species (see Source Data). The selected genome sequences were downloaded from Refseq database. The protein sequences of the genomes were produced using Prokka v1.14.6[113]. The Pdu proteins of *S.* Typhimurium strain LT2 were used as references, including PduA, PduB, PduD, PduJ, PduK, PduM, PduN, PduU, and PduX. The Pdu protein sequences were queried against the proteomes using BLASTp v2.5.0+ [112] with a threshold of 90% amino acid identity. The similarity of the proteins was summarized from the BLAST results.

To build the phylogenetic tree of the 61 bacterial genomes, the universal single-copy orthologs of Gammaproteobacteria were obtained from each genome using BUSCO v5.2.2[114]. *Vibrio cholerae* MS6 was used as an outgroup to root the phylogenetic tree. From the BUSCO result, 327 genes were found to be present in all the genomes. Each gene set was aligned with Mafft v7.475[115]. The alignments were concatenated with SeqKit v0.15.0[116], then trimmed with Trimal v1.4[117] with the automatic method. A phylogenetic tree was constructed from the alignment with Fasttree v2.1.10[118] using the JTT + Gamma model.

The protein sequences of PduB1-37, PduM, and PduK were extracted from *S.* Typhimurium LT2, *Klebsiella pneumoniae* HS11286, *Escherichia coli* O104:H21 strain ATCC_BAA-178, and *Citrobacter freundii* ATCC_8090 as representatives of each genus. The structure of each protein was predicted by AlphaFold2[119], accessed via ColabFold[120]. The structures were visualized in ChimeraX1.3[121]. The protein structures were colored by the conservation value of each residue in alignment with the proteins from the 61 isolates. The conservation values were automatically calculated by ChimeraX1.3 using the entropy-based measure from AL2CO[122].

**Statistics and reproducibility**. For confocal microscopy and TEM imaging, at least three biologically independent experiments were repeated. Representative images were shown in this paper.

**Reporting summary**. Further information on research design is available in the Nature Research Reporting Summary linked to this article.

## Data availability

All data needed to evaluate the conclusions in the paper are present in the main text or the supplementary materials. The complete bacterial genomes were downloaded from the Refseq bacterial database (according to https://www.ncbi.nlm.nih.gov/genome/doc/ftpfaq/).

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

## Acknowledgements

We thank the Liverpool Centre for Cell Imaging and Biomedical Electron Microscopy Unit for providing technical assistance and provision for microscopic imaging. This work was supported by the National Key R&D Program of China (2021YFA0909600), the National Natural Science Foundation of China (32070109), the Royal Society (URF\R\180030, UF120411, RGF\EA\181061, and RGF\EA\180233 to L.-N.L.), the Biotechnology and Biological Sciences Research Council (BBSRC) (BB/V009729/1, BB/M024202/1, and BB/R003890/1 to L.-N.L.), the Leverhulme Trust (RPG-2021-286 to L.-N.L.), the China Scholarship Council (to M.Y.) and, in part, by a Wellcome Trust Senior Investigator Award (grant number 106914/Z/15/Z to J.C.D.H.). For the purpose of open access, the authors have applied a CC BY public copyright license to any Author Accepted Manuscript version arising from this submission.

## Author contributions

M.Y., N.W., G.F.D., Y.L., X.Z., F.H., and Y.S. performed research; M.Y. and L.-N.L. analyzed data; M.Y., J.C.D.H., and L.-N.L. designed research and wrote the article with contributions from other authors.

## Competing interests

The authors declare no competing interests.
