## [Peer Review File · Nature Communications]

Reviewer comments, first round review –

Reviewer #1 (Remarks to the Author):

General comments: The work provided by the authors is a substantial and impactful contribution to the field of bacterial microcompartment biology. The results are well controlled, persuasive, and provide detailed information into the mechanism of metabolosome biogenesis, findings that will be useful to those seeking to re-engineer these organelles for non-native purposes. A number of novel discoveries, such as a putative function of PduM, are proposed and supported by the data. However, the interpretation and discussion of the findings can be strengthened significantly. Specifically, some findings made in the manuscript have alternative explanations that may be accurate, in light of existing literature. This is not to discredit the findings made by the authors or to imply that they are unimportant or unsubstantiated – indeed, these are interesting and useful findings! However, proper context and alternative explanations should be provided given that these results are in disagreement with a number of published works. Therefore, outlined below are suggestions for improving the manuscript.

Major comments:

- 1) Data on conservation is misleading, as only *Salmonella* isolates were analyzed. The expected result for these isolates is high conservation and a >90% conservation is neither surprising, nor informative. This is true for both the PduB N-terminus findings and PduM findings. This can be addressed in the following ways:
 - a) Ideally, the analysis should be expanded to other genera (eg *Citrobacter*, *Klebsiella*).
 - b) Please provide the context of these conservation scores. How does the conservation of the PduB N-terminus compare to other essential Pdu MCP components (i.e. PduA, PduJ, PduN, and PduD)? Is it more or less conserved than other regions of the operon, especially those without clear functional roles (such as PduU or PduX)?
 - c) For PduM, provide context relative to other genes. Is it present more or less frequently than other genes in the operon?
 - d) The phylogenetic trees should contain an outgroup for comparison
 - e) Please adjust the interpretation of the results to accurately represent the findings in light of the above context. If the N-terminus is as conserved as other parts of the operon, say “as expected, the N-terminus of PduB is conserved to a similar degree as other essential Pdu MCP genes/regions.” Adjust the interpretations of the PduM results similarly.
- 2) The finding that PduM deletion leads to malformed MCP-like structures has been reported previously, so please cite this finding and indicate that your finding replicates it (Sinha, JMB, 2011). It is also important to note that the observed elongated structures are very likely due to a polar effect disrupting expression of PduN, which as the findings in this manuscript indicate, can lead to elongated structures when it is knocked out. In fact, studies indicate that genetic disruptions as far away as the PduL locus can have similar effects (Nichols, BEJ, 2020). Please note these findings as potential explanations for the observed results while citing prior literature about the importance of PduN.
- 3) The finding that PduJ knockout strains form polar bodies is not corroborated by the literature, as studies have indicated this leads to elongated structures (Cheng, J. Bact., 2010) or no change in MCP assembly (Kennedy, JMB, 2021). Interestingly, the PduJ and PduK knockouts in this manuscript look

very similar, indicating that the PduJ knockout may be causing a polar effect that disrupts PduK expression. Please soften the finding to include this possibility, while also noting that this does not entirely agree with literature findings (and provide possible reasons as to why your results may differ). Ideally, these experiments should be rerun with knockouts that are more minimal and less susceptible to such polar effects.

Similarly, in the introduction, you note that “PduBB’, PduJ and PduN are the shell components that are essential for Pdu BMC formation and structure.” (Lines 56-57); however, as noted above, literature has shown that either PduA or PduJ is essential, not exclusively PduJ. Please modify this statement accordingly.

4) The claim that PduK is important for subcellular microcompartment distribution is the same as that made in ref (22), in which polar aggregates are observed by thin cell section EM. The fluorescence microscopy (and similar TEM images) and plasmid supplement data provides similar evidence for PduK being important for either subcellular distribution or shell formation; however, in the absence of TEM on purified compartments, whether these polar aggregates are well-formed compartments or aggregated shell/cargo proteins is not clear, and the possibility of either conclusion in the absence of further data should be noted.

5) Given the high information density in each figure, it would be helpful to label each set of microscopy images not only with which protein is labeled with which color, but also what the strain background is for each set of images, perhaps centered in a larger font above the panel (ie, $\Delta pduB1-37$ for Fig 3b) or by visually indicating the knockout in the operon. This would make it significantly easier to interpret the images without the need to reference the caption.

6) Information on plasmid supplement assays is incomplete, as the induction conditions and details of the plasmid used are not provided. For reproducibility, please include:

- a) Details on pXG10 plasmid—Either a reference to a full plasmid map or key information about the plasmid (the origin of replication, antibiotic resistance cassette, and promoter).
- b) Induction conditions (time of induction, inducer concentration)

7) While the green and red labels and micrograph colorings are useful in that they differentiate between mCherry and GFP, they are not red-green colorblind friendly. Please alter to pseudocoloring and labels that are colorblind friendly.

8) Minor comment – Line 392; PduJ can also form nanotubes.

9) Minor comment – Line 16; “Protein peptides” is redundant

10) Minor comment – Line 31; “Membrane” can refer to protein or lipid barriers; “lipid” would be more appropriate here.

Reviewer #2 (Remarks to the Author):

The manuscript under review, authored by Yang et al. presents a model for Pdu metabolosome biogenesis in *S. typhimurium* LT2. The methods used include genetic analysis, live-cell fluorescence, electron microscopy, and growth assays. The main findings proposed by the authors are: (1) The Pdu metabolosome undertakes both “shell first” and “cargo first” assembly pathways, (2) shell and cargo

assemblies occur independently at the cell poles, (3) the internal cargo core is formed through ordered assembly of multiple enzyme complexes, (4) and the internal cargo core exhibits liquid-like properties within the metabolosome architecture. Additionally, the authors report findings that suggest the PduB N-terminus and PduM are necessary for binding between the shell and cargo assemblies and that PduK has an important role for spatial localization of the Pdu metabolosomes and their movement from the poles to other locations within the cell.

The article gives an interesting and detailed analysis of the biogenesis of the Pdu metabolosome and the different protein components that make up the metabolosome using knockout mutants and live-cell fluorescent imaging. The authors' thorough analysis supports and synthesizes previous findings on the specific roles of certain Pdu proteins.

Overarching considerations include:

- 1) The appropriateness, as written, for the Nature Communications readership—the study is intricately detailed and requires much related reading, where one finds many of the findings already presented piecemeal by mutational analysis in work beginning 30 years ago. It was difficult to follow because of the need to continually refer back to the understand protein was being discussed. I would expect to be even more challenging for any one not well-versed in the MCP field.
- 2) The key conclusions—that the PDU BMC uses both shell and core first (which sometimes elides into concomitant) is somewhat unsatisfactory (please see below). The second major conclusion, that this is true for metabolosomes in general, is questionable. Two key components, the PduB and PduM are proteins with features found only (only with very few exceptions) in the PDU MCP.
- 3) The authors neglect previous work on encapsulation peptides by many groups—Key papers include Juodeikis et al. *MicrobiologyOpen* 2020, Lee et al., *Metabolic engineering* 2016, Erbilgin et al., *PLoS Biol* 2016, Kinney et al., *JBC* 2012, Jakobsen *Protein Science*). These studies need to be contextualized here as EPs are indeed universal among metabolosomes and likely play a role that should be better accounted for in the authors' model.
- 4) Despite neglecting consideration of the possibility of artifacts introduced by the tagging, I find this an important contribution, consisting of a heroic amount of work that is well done, and likely valuable to the PDU MCP community as a comprehensive synthesis of the roles of PDU proteins.

Major Points:

a. Fig5D and lines 305-307: Authors should quantify the frequency of colocalized shell and core events within the same cell at the initial time point of 1 hour. From the example images in Fig.5D, it appears colocalization of shell and core aggregates/assemblies also occurs, implying some frequency of concomitant assembly in addition to the individual core-only or shell-only events. This analysis should then be reconciled with the proposed pathway for assembly in relation to the core-first or concomitant pathways proposed for beta and alpha carboxysomes, respectively, as the difference between concomitant assembly and the model proposed here is not clear. Moreover, in the context of the proposed PDU assembly pathway, it's not clear how the authors envision the evolution of shell first and core first events leading to the assembly of the mature PDU organelle. They invoke the idea of cargo "import" following shell assembly as described on lines 308-311 (see quote below, bold is mine), but this hypothesis should be unpacked for the reader – do the authors propose that fully contiguous empty shells disassemble somehow to incorporate large enzyme cargo assemblies? See lines 308-311:

"The colocalization of shell and cargo fluorescence was then visualized after 1.5h induction with 1,2-PD, indicating the subsequent import of cargo enzymes and encapsulation of shell structures,

respectively, prior to the formation of the Pdu BMCs (Fig. 5D).”

b. The authors nicely contextualize the conservation of key PDU elements among *Salmonella* strains, however, many of these shell/core proteins are not conserved among other BMC types (other BMCs encapsulating different metabolic segments, or even other PDU subtypes) which significantly limits the extrapolation of these results to other systems. The authors could strengthen this assertion with some of the findings on the alpha carboxysome by the Cannon-Heinhorst group (e.g. Life 5:1141-1171, 2015) where alpha carboxysome-specific proteins drive the assembly and both empty shells and partial carboxysomes are observed.

c. I think the FRAP experiment in Fig.5 is a clever way to probe dynamics of the PDU core, but given the large variability in the Pearson's coefficient for the Δ PduN (Fig.2D) strain, it would be a useful control to measure the FRAP of PduCDE-sfGFP in the absence of PDU induction (no 1,2 PD) to compare the mobility of the complex in solution versus “inside” the elongated PDU organelles (which may or may not be truly enclosed structures).

d. Lines 160-170: Although the authors briefly reference structural defects (or absence of a shell) as a possible explanation of the growth enhancement results of Fig. 2D, I believe the authors should further clarify their explanation of the improved growth rate in strains with shell protein deletions. By “substrate availability” are the authors referring to the B12 cofactor, as I believe was used to rationalize the similar growth enhancements observed in the cited works? If so, the use of ‘permeability’ in this paragraph is vague and potentially misleading, as the permeability being measured is likely not to the substrate (1,2 PD) (or intermediates/products) of the metabolic segment inside the BMC. Instead, it seems the changes in ‘permeability’ that lead to an increased growth rate are being explained as the result of substantial structural compromise to the shell that abolishes its ability to act as a diffusive barrier because it is no longer intact, therefore allowing access of apo-PduCDE to the large B12 molecule that would otherwise be limiting to enzymes encapsulated inside a contiguous shell; however if this is the case, why are no deleterious effects of aldehyde leakage also observed for these deletions?

e. In observing the growth rates of the Pdu mutants compared to the WT, results showed increased growth rates for most of the mutants. This is suggested to be due to formation of the enzymatic core without encapsulation, allowing for greater access to environmental B12. This seems the likely case but wouldn't this also release cytotoxic intermediates that would damage the cell? Would there be any way to quantify increased concentrations of these intermediates and compare them to the WT?

f. Supplementary data with alignments / conservation: the sequences are compared among very similar *Salmonella* strains so a high conservation of amino acids is expected; are the pdu-associated sequences more conserved than expected for such similar strains?

g. potential effects of fluorescent labeling technique are not discussed (e.g. fluorescently tagged proteins might be insoluble and form inclusion bodies at poles)

Minor comments

a. Lines 136-137: the authors should provide fluorescence images for the WT cells expressing PduJ-sfGFP and PduE-mCherry as a control for the Δ PduA strain images in Fig2A.

b. Lines 142-143: to avoid confusion, it would be best to qualify (as done a few lines above) that PduJ-sfGFP was used for the Δ PduA strain.

c. Line 182: authors should specify that this could also be the reasonable result of fusion with the fluorophore.

d. It wasn't initially clear that the PduCDE enzyme complexes in Fig3D were expressed from a plasmid and which subunit was tagged with sfGFP (although my assumption is PduE). I suggest

adding this detail in the figure legend to avoid any confusion. Related to this point, the main text referring to this panel would also benefit from some elaboration that makes clear the difference between the fluorescence phenotypes observed here when PduCDE, with a fluorophore tagged subunit, is expressed from a plasmid (if my interpretation is correct) as opposed to only one of the subunits (i.e. PduE-sfGFP shown in previous figures) that lack the PduC and PduD partners encoded only in the genome and not expressed due to the lack of 1,2PD.

e. Fig5C “C” label overlapping with figure panel. Fig 5E: “pole” and “Midcell” labels overlapping with boundary box.

f. Lines 455-457: Authors refer to the PDU encapsulation peptides as a “model system of the assessment of the LLPS-mediated ordered assembly of protein organelles”, however the authors should clarify that encapsulation peptides may not meet the definition of conventional LLPS-mediated systems, as they have defined them, in part, by their secondary structure – an amphipathic alpha helix. References to EP literature and clarification/discussion of how EPs and characterized LLPS systems differ is warranted.

g. Lines 488-490: authors should describe how these small volume (100 ul) cultures were grown and OD's recorded – was it using the same plate reader system described later?

h. Line 527: reference for the Gibson assembly method should be added:

Daniel G. Gibson et al., “Enzymatic Assembly of DNA Molecules up to Several Hundred Kilobases,” *Nature Methods* 6, no. 5 (May 2009): 343–45, <https://doi.org/10.1038/nmeth.1318>.

i. In the Pearson's R plots in the SI material, not all lines extend from the origin (e.g. S17 PduU-PduA), nor do all scatter plots appear to be symmetrical about the line. For example, Fig. S10 PduP-PduE is calculated as having a higher coefficient but doesn't appear to be a better 'fit' to the white line than the PduK-PduA. How do these examples impact interpretation of “colocalization”?

line 236: Should those refer to Figure 4 instead of 2?

line 368: grammar, "assembly" should be "assemble"

line 934 / Figure 1: PduV is missing in list of different components, PduM is listed as shell protein but doesn't have a 00936/03319 pfam

line 958 / Figure 2: error for the Δ pduA measurement seems higher than the 0.02 when comparing to errors of other plots

line 966 / Figure 3: panel H: make it clear that those two images are not from the same strain, e.g. Δ pduB1-37 vs Δ pduB1-37 WITH expression of fluorescently labeled proteins

line 1000 / Figure 4: panel H: as with Figure 3, those strains differ since the TEM image is not from the strain with fluorescently marked proteins

Reviewer #3 (Remarks to the Author):

The study presented by Liu group provides an in-depth study, where they have individually characterized the protein-protein interaction and assembly involved in catabolic bacterial microcompartment formation. Liu group in several of their earlier studies successfully utilized microscopy as an effective tool to study the complex traits of microcompartment formation. In this study, using fluorescent labeled individual BMC shell protein and an internal protein they answer one simple question “Which one happens first: the egg or the chicken?” or in the context of the study, whether the core forms first or the shell.

While several earlier studies specifically from Bobik group has forwarded substantial knowledge (as also presented here), finally allowed this study to use all these fragmented knowledges into one single study. However, I am a bit lost in too many microscopic pictures and where to find the real

question. While I can understand one key point in this study is the role of PduM, this protein however is not present in all BMCs (reported till date), hence the function of PduM cannot be universal. The study thus is not focused and may not appeal to a broad audience for a journal like *Nar. Comm.*

Nonetheless, the study is interesting to people working in this field but needs to be improved:

1. Line 17: At the onset calling BMCs as organelles might not be correct.
2. Line 30: Which kind of cells?
3. Line 47: which are examples of bacteria in anoxic niches? All those cited are only reviews.
4. Line 127: The microscopy was done 1hr post induction with 1,2-PD, how do the cells look at the mid exponential phase?
5. Line 141: the signals are in discrete patches; do they remain the same after 15 h (~ 0.5 OD)?
6. In Figure 2B: Δ pduM, in this strain PduB is in full length still PduE m-cherry donot co-localize? Is there a specific reason?
7. Line 147-150: Pdu J is not absolutely necessary for Pdu BMC formation since there are bacteria that do not have PduJ or this must only be in case of *S. enterica*. For the same reason, it is important to see the fluorescent signals within the cells at a later time point.
8. Line 157: In Δ pduB1-37, if the assembly of shell and cargo (PduE) is spatially separated, how do the cells show higher growth phenotype? In the bioenergetic point of view, a faster growth phenotype is in direct correlation to its ATP turnover. Since the cells grow solely on 1,2-PD, propionate formation is likely the only energy conserving mechanism. A spatially separated PduCDE assembly outside BMC likely results propionaldehyde to be generated outside BMC which is supposed to be toxic to the cells. In the same argument, if we consider lowering VitB12 concentration results in the phenomenon, bioenergetically this is not possible (Fig 2D).
9. Line 181-182: The insoluble theory might not be true: it is likely PduO and PduP is constitutively expressed
10. If we take figure 2B Δ pduB1-37 and correlate figure 3B, this undoubtedly shows the BMC and the internal core protein localization is completely aloof. So, in this case, purification of BMC will result in purified shell, Is this true?
11. Line 238, 248-249: This might be true only with *Salmonella*, not all BMC clusters have PduM. In our experience, PduM is not absolute and intact BMCs are well formed without it.
12. Line 265: in Δ pduB1-37/ Δ pduK, fig4F shows PduB' not to colocalize with PduA rather to be dispersed over the cell, am I correct?
13. Fig 5. I am not an expert in fluorescence microscopy and hence cannot interpret most of the observed results and how relevant is it in current context. The results are dependent on several factors like style of experiment, time at which the results recorded. I am also not sure how tagging a GFP (27kda) protein to PduA (9kda) protein allows BMC formation (as I can gauge a 9kda protein will not interact similarly to its partner once it is modified to a 37kda protein).
14. Line 320-322: figure 3B actually shows core and BMC are separate as in my earlier comment. If core is separate, how can it be within BMC?
15. Figure 6b: if I am not wrong: PduE is supposed to have an N-terminal encapsulation peptide and hence should localize similar to PduL or at least encapsulated within BMC (as per fluorescence signal is considered). Not having a discrete signal hints an unfolded protein or something else.
16. Fig6F: Don't know how to interpret the data

RESPONSES TO REVIEWERS' COMMENTS

Reviewer #1:

General comments: The work provided by the authors is a substantial and impactful contribution to the field of bacterial microcompartment biology. The results are well controlled, persuasive, and provide detailed information into the mechanism of metabolosome biogenesis, findings that will be useful to those seeking to re-engineer these organelles for non-native purposes. A number of novel discoveries, such as a putative function of PduM, are proposed and supported by the data. However, the interpretation and discussion of the findings can be strengthened significantly. Specifically, some findings made in the manuscript have alternative explanations that may be accurate, in light of existing literature. This is not to discredit the findings made by the authors or to imply that they are unimportant or unsubstantiated – indeed, these are interesting and useful findings! However, proper context and alternative explanations should be provided given that these results are in disagreement with a number of published works. Therefore, outlined below are suggestions for improving the manuscript.

Major comments:

1) Data on conservation is misleading, as only Salmonella isolates were analyzed. The expected result for these isolates is high conservation and a >90% conservation is neither surprising, nor informative. This is true for both the PduB N-terminus findings and PduM findings. This can be addressed in the following ways:

- a. Ideally, the analysis should be expanded to other genera (eg Citrobacter, Klebsiella).*
- b. Please provide the context of these conservation scores. How does the conservation of the PduB N-terminus compare to other essential Pdu MCP components (i.e. PduA, PduJ, PduN, and PduD)? Is it more or less conserved than other regions of the operon, especially those without clear functional roles (such as PduU or PduX)?*
- c. For PduM, provide context relative to other genes. Is it present more or less frequently than other genes in the operon?*
- d. The phylogenetic trees should contain an outgroup for comparison*
- e. Please adjust the interpretation of the results to accurately represent the findings in light of the above context. If the N-terminus is as conserved as other parts of the operon, say “as expected, the N-terminus of PduB is conserved to a similar degree as other essential Pdu MCP genes/regions.” Adjust the interpretations of the PduM results similarly.*

Reply: We agree with the reviewer's comments. We have expanded the analysis to other genera, including *Klebsiella*, *Escherichia*, and *Citrobacter*. The new results are shown in Fig. S10 in the revised manuscript. Fig. S10 includes the conservation scores of other Pdu components, as a comparison with PduB¹⁻³⁷, PduK, and PduM. We established that the level of conservation of PduB¹⁻³⁷ is mirrored by the conservation of other structural or essential Pdu proteins (PduA/B/D/J/N/U).

We added an outgroup (*Vibrio cholerae*) to generate a rooted phylogeny (Fig. S10). The relevant interpretations have been added to the revised manuscript. In addition, we have added protein-structure prediction results that illustrate the high structural similarity of PduB¹⁻³⁷, PduK, and PduM proteins (Fig. S10).

2) The finding that PduM deletion leads to malformed MCP-like structures has been reported previously, so please cite this finding and indicate that your finding replicates it (Sinha, JMB, 2011).

Reply: Thanks for pointing this out. We have cited Sinha, JMB, 2011 in support of our observations in the revised manuscript.

It is also important to note that the observed elongated structures are very likely due to a polar effect disrupting expression of PduN, which as the findings in this manuscript indicate, can lead to elongated structures when it is knocked out. In fact, studies indicate that genetic disruptions as far away as the PduL locus can have similar effects (Nichols, BEJ, 2020). Please note these findings as potential explanations for the observed results while citing prior literature about the importance of PduN.

Reply: In this study, we used the Lambda Red method (Datsenko and Wanner, *PNAS*, 2000, 97: 6640-6645) to generate gene-deletion mutants using an approach designed to avoid polar effects. The antibiotic resistance cassette was removed using the FLP recombinase expressing plasmid pCP20. Each deletion replaced the coding sequence with a short sequence that did not disrupt the reading frame (confirmed by gene sequencing) or translational coupling. Previously, we used RNA-seq to assess transcription in multiple mutants constructed using a similar approach and determined that transcription downstream of 18 "clean" deletion mutants in *S. Typhimurium* was not affected by the mutation (Figure 1 in Colgan et al., *PLoS Genetics* 2016, 12(8): e1006258).

To add independent confirmation of our findings, we generated a 'scarless' *pduM* deletion mutants by genome editing (Martínez-García and de Lorenzo, *Environ. Microbiol.* 2011, 13: 2702-2716), and observed a similar fluorescence profile in the *pduM* scarless mutation (Fi. R1) compared with that in the $\Delta pduM$ generated by Lambda Red method.

Taken together, our data show that the observed elongated structures are not due to a polar effect that disrupted expression of PduN.

Fig. R1. Confocal images of the *pduM* deletion mutant shown in the manuscript (left, Fig. 2b) and the new scarless *pduM* deletion mutant (right) expressing PduE mCherry/PduA-sfGFP grown in MIM+1,2-PD media.

3) *The finding that PduJ knockout strains form polar bodies is not corroborated by the literature, as studies have indicated this leads to elongated structures (Cheng, J. Bact., 2010) or no change in MCP assembly (Kennedy, JMB, 2021).*

Reply: Our observations that the *pduJ* deletion resulted in the formation of polar aggregates and occasionally elongated Pdu BMC structures were consistent with previous work (Cheng et al., *J. Bacteriol.* 2011, 193: 1385-1392) that we have cited in the manuscript. This paper by Cheng et al. 2021 reported that 22% of cells had highly elongated structures, while ~20% of the examined cells contained amorphous inclusion bodies. To clarify this point, we have repeated the imaging of the $\Delta pduJ$ strain and have highlighted the elongated structures with arrows (Fig. 2b).

Interestingly, the PduJ and PduK knockouts in this manuscript look very similar, indicating that the PduJ knockout may be causing a polar effect that disrupts PduK expression. Please soften the finding to include this possibility, while also noting that this does not entirely agree with literature findings (and provide possible reasons as to why your results may differ). Ideally, these experiments should be rerun with knockouts that are more minimal and less susceptible to such polar effects.

Reply: As explained in the response to Comment 2) above, the gene-deletion method that we used was designed to avoid polar effects. To confirm the function of PduJ, we have generated a 'scarless' *pduJ*-deletion mutant (Fig. R2). This 'scarless' mutant produced similar signals observed with the $\Delta pduJ$ generated by the Lambda Red method in this manuscript.

Fig. R2. Confocal images of the scarless *pduJ* deletion mutant expressing PduE mCherry/PduA-sfGFP grown in MIM+1,2-PD media. For comparison, see the images generated with *pduJ*-deletion mutant in the manuscript (Fig. 2b).

Similarly, in the introduction, you note that “PduBB’, PduJ and PduN are the shell components that are essential for Pdu BMC formation and structure.” (Lines 56-57); however, as noted above, literature has shown that either PduA or PduJ is essential, not exclusively PduJ. Please modify this statement accordingly.

Reply: We have modified the statement in introduction to ‘PduBB’, PduA or PduJ, and PduN are the shell components that are essential for Pdu BMC formation and structure’. Relevant literatures have been cited accordingly.

4) The claim that PduK is important for subcellular microcompartment distribution is the same as that made in ref (22), in which polar aggregates are observed by thin cell section EM. The fluorescence microscopy (and similar TEM images) and plasmid supplement data provides similar evidence for PduK being important for either subcellular distribution or shell formation; however, in the absence of TEM on purified compartments, whether these polar aggregates are well-formed compartments or aggregated shell/cargo proteins is not clear, and the possibility of either conclusion in the absence of further data should be noted.

Reply: To address this comment, we isolated the Pdu MCPs from $\Delta pduK$, and checked them by SDS-PAGE and negative staining TEM (Fig. S12 in the revised manuscript). The results confirmed that the polar aggregates observed were well-formed microcompartments. We have also cited ref (27) accordingly.

5) Given the high information density in each figure, it would be helpful to label each set of microscopy images not only with which protein is labeled with which color, but also what the strain background is for each set of images, perhaps centered in a larger font above the panel (ie, $\Delta pduB1-37$ for Fig 3b) or by visually indicating the knockout in the operon. This would make it significantly easier to interpret the images without the need to reference the caption.

Reply: We are grateful for the review’s excellent suggestion. Accordingly, we have labelled the strain background above the panel for Figs. 3 and 4.

6) Information on plasmid supplement assays is incomplete, as the induction conditions and details of the plasmid used are not provided. For reproducibility, please include:

- a) Details on pXG10 plasmid—Either a reference to a full plasmid map or key information about the plasmid (the origin of replication, antibiotic resistance cassette, and promoter).
- b) Induction conditions (time of induction, inducer concentration)

Reply: We have added the details and reference related to the pXG10 plasmid in Table S1 in the revised manuscript. The ssDNA oligos used in complementation experiments have been added to Table S2. We have modified the plasmid and induction conditions in Methods.

7) While the green and red labels and micrograph colorings are useful in that they differentiate between mCherry and GFP, they are not red-green colorblind friendly. Please alter to pseudocoloring and labels that are colorblind friendly.

Reply: We choose to use the combination of green and red because this complementary colour pair provides the highest contrast that allow the fluorescence signals to be distinguished explicitly, in particularly for less abundant proteins in the case of BMCs that have a wide range of protein abundance (Yang et al., *Nat Commun*, 2020, 11(1): 1976). As shown in Fig. R3 below, we have compared different complementary colour pairs (green-red, cyan-magenta, cyan-yellow), and it is apparent that green and red offer the best contrast for colocalization analysis.

For the same reason, the green-red pair has been commonly used in publications for fluorescent imaging and colocalization analysis, including the study of carboxysome biogenesis in cyanobacteria (Linsley, et al. *Nat Commun* 2021, 12: 1-14; Gabriel, Christian, et al. *Nat Commun* 2021, 12: 1-15; Cameron, Jeffrey C., et al. *Cell* 2013, 155: 1131-1140).

We are aware that an estimated 8% of males and less than 1% of females have the Deuteranomaly or Protanomaly forms of colourblindness. However, given that our manuscript includes substantial numbers of confocal fluorescence images of Pdu MCP proteins that have a wide range of abundance, there is no alternative to the green and red colourings used in our manuscript.

Fig. R3. Comparison of the confocal images using different colour scheme combinations. The confocal images of PduE-mCherry/PduA-sfGFP in LT2-WT was used as an example.

8) *Minor comment – Line 392; PduJ can also form nanotubes.*

Reply: We have changed the text to “PduA, PduB, and PduJ can form nanotube shell structures”.

9) *Minor comment – Line 16; “Protein peptides” is redundant.*

Reply: We have deleted “peptides”.

10) *Minor comment – Line 31; “Membrane” can refer to protein or lipid barriers; “lipid” would be more appropriate here.*

Reply: We have replaced “membrane” with “lipid”.

Reviewer #2:

The manuscript under review, authored by Yang et al. presents a model for Pdu metabolosome biogenesis in S. typhimurium LT2. The methods used include genetic analysis, live-cell fluorescence, electron microscopy, and growth assays. The main findings proposed by the authors are: (1) The Pdu metabolosome undertakes both “shell first” and “cargo first” assembly pathways, (2) shell and cargo assemblies occur independently at the cell poles, (3) the internal cargo core is formed through ordered assembly of multiple enzyme complexes, and (4) the internal cargo core exhibits liquid-like properties within the metabolosome architecture. Additionally, the authors report findings that suggest the PduB N-terminus and PduM are necessary for binding between the shell and cargo assemblies and that PduK has an important role for spatial localization of the Pdu metabolosomes and their movement from the poles to other locations within the cell. The article gives an interesting and detailed analysis of the biogenesis of the Pdu metabolosome and the different protein components that make up the metabolosome using knockout mutants and live-cell fluorescent imaging. The authors’ thorough analysis supports and synthesizes previous findings on the specific roles of certain Pdu proteins.

Overarching considerations include:

1) The appropriateness, as written, for the Nature Communications readership—the study is intricately detailed and requires much related reading, where one finds many of the findings already presented piecemeal by mutational analysis in work beginning 30 years ago. It was difficult to follow because of the need to continually refer back to the understand protein was being discussed. I would expect to be even more challenging for any one not well-versed in the MCP field.

Reply: Numerous studies have used genetic deletion, thin-section EM, or biochemical analysis to explore the functions of individual proteins in the assembly of Pdu BMCs. However, a comprehensive understanding of the assembly pathway of Pdu BMCs is lacking. In this study, we developed a system to express fluorescently tagged Pdu proteins and applied state-of-the-art live-cell fluorescence imaging, together with genetic analysis, EM and growth assays, to perform a systematic characterization of the stepwise biogenesis of Pdu BMCs in the model organism *S. typhimurium* LT2. The study has provided new insight into the the assembly principles of Pdu BMCs that have not been well characterized previously (as the reviewer summarized above).

2) The key conclusions—that the PDU BMC uses both shell and core first (which sometimes elides into concomitant) is somewhat unsatisfactory (please see below). The second major conclusion, that this is true for metabolosomes in general, is questionable. Two key components, the PduB and PduM are proteins with features found only (only with very few exceptions) in the PDU MCP.

Reply: We have added further image analysis and clarified the statements related to our “shell and core first” conclusion. Please see our response to major point ‘a’ below.

Current knowledge about the assembly of diverse BMCs has highlighted the requirement for linker proteins to mediate the BMC shell-cargo association, such as CsoS2 in alpha-carboxysomes, CcmM and CcmN in beta-carboxysomes, and PduB and PduM in Pdu BMCs (in this work). Unlike carboxysomes, bacterial metabolosomes (including Pdu BMCs) sequester a series of cargo enzymes.

Our study reports the discovery that the biogenesis of Pdu BMCs adopts both “Shell first” and “Cargo first” pathways that occur independently; the cargo enzymes are prone to assemble together, even though several of the cargos contain their own encapsulation peptides, to generate an internal enzymatic core that possesses a liquid-like status.

We propose that such mechanisms are common in metabolosomes and merit further investigation. However, the linker proteins and encapsulation peptides may vary depending on the types of metabolosomes. We have added relevant discussion to the revised manuscript.

3) The authors neglect previous work on encapsulation peptides by many groups—Key papers include Juodeikis et al. MicrobiologyOpen 2020, Lee et al., Metabolic engineering 2016, Erbilgin et al., PloS Biol 2016, Kinney et al., JBC 2012, Jakobsen Protein Science 2017). These studies need to be contextualized here as EPs are indeed universal among metabolosomes and likely play a role that should be better accounted for in the authors’ model.

Reply: Thanks for pointing this out. We have added the relevant references to the revised manuscript.

4) *Despite neglecting consideration of the possibility of artifacts introduced by the tagging, I find this an important contribution, consisting of a heroic amount of work that is well done, and likely valuable to the PDU MCP community as a comprehensive synthesis of the roles of PDU proteins.*

Reply: The effects of the fluorescent labelling technique can be found in our detailed response to major points ‘g’ below. To confirm the fluorescence imaging observations, we have performed extensive studies using thin-section EM on cells without fluorescence tagging. We have modified the relevant statements in the revised manuscript.

Major Points:

a. *Fig5D and lines 305-307: Authors should quantify the frequency of colocalized shell and core events within the same cell at the initial time point of 1 hour. From the example images in Fig.5D, it appears colocalization of shell and core aggregates/assemblies also occurs, implying some frequency of concomitant assembly in addition to the individual core-only or shell-only events. This analysis should then be reconciled with the proposed pathway for assembly in relation to the core-first or concomitant pathways proposed for beta and alpha carboxysomes, respectively, as the difference between concomitant assembly and the model proposed here is not clear. Moreover, in the context of the proposed PDU assembly pathway, it’s not clear how the authors envision the evolution of shell first and core first events leading to the assembly of the mature PDU organelle. They invoke the idea of cargo “import” following shell assembly as described on lines 308-311 (see quote below, bold is mine), but this hypothesis should be unpacked for the reader – do the authors propose that fully contiguous empty shells disassemble somehow to incorporate large enzyme cargo assemblies? See lines 308-311: “The colocalization of shell and cargo fluorescence was then visualized after 1.5h induction with 1,2-PD, indicating the subsequent import of cargo enzymes and encapsulation of shell structures, respectively, prior to the formation of the Pdu BMCs (Fig. 5D).”*

Reply: As the reviewer suggested, we have quantified the frequency of colocalized shell and cargo events (indicating the concomitant assembly) that were observed within the same cell at the 1 h time point (Table S3), and have added relevant descriptions to the revised manuscript.

Please note that to reduce the photobleaching that would be caused by continuous microscopic scanning, the confocal fluorescence images were captured at the 1 h and 1.5 h time points. The formation of cargo and shell assemblies was observed roughly at 1 h, but the actual assembly may have occurred earlier. For the spots with colocalized green and red signals, we cannot exclude the possibility that they represent a subsequent stage after the shell or cargo first assembled independently and then associated with cargo or shell proteins, respectively. Thus, it is possible that the initial Shell-first or Cargo-first events were missed by the confocal imaging.

Regarding the evolution of shell-first and core-first events leading to the assembly of the mature Pdu BMC, we hypothesize that the initially formed shell or cargo assemblies are partially assembled. This would allow the partially assembled shells to recruit cargo enzymes without the need to disassemble. Such a mechanism is reminiscent of the partially formed alpha-carboxysomes observed previously. We have revised the relevant statements in Discussion.

b. *The authors nicely contextualize the conservation of key PDU elements among Salmonella strains, however, many of these shell/core proteins are not conserved among other BMC types (other BMCs encapsulating different metabolic segments, or even other PDU subtypes) which significantly limits the extrapolation of these results to other systems. The authors could strengthen this assertion with some of the findings on the alpha carboxysome by the Cannon-Heinhorst group (e.g. Life 5:1141-1171, 2015) where alpha carboxysome-specific proteins drive the assembly and both empty shells and partial carboxysomes are observed.*

Reply: Our updated bioinformatic results (Fig. S10) have revealed the conservation of Pdu proteins in diverse Pdu BMC-containing bacterial genera including *Klebsiella*, *Escherichia*, *Citrobacter*, and *Salmonella*. The key Pdu elements identified in this study may be not conserved in other bacterial metabolosomes, but similar construction principles may be applied and functionally similar proteins

in other metabolosomes remain to be investigated. We have added the reference (*Life* 2015, 5: 1141-1171) to the text where we compared the assembly pathway of Pdu BMCs with carboxysomes in Discussion.

c. I think the FRAP experiment in Fig.5 is a clever way to probe dynamics of the PDU core, but given the large variability in the Pearson's coefficient for the Δ PduN (Fig.2D) strain, it would be a useful control to measure the FRAP of PduCDE-sfGFP in the absence of PDU induction (no 1,2 PD) to compare the mobility of the complex in solution versus "inside" the elongated PDU organelles (which may or may not be truly enclosed structures).

Reply: We are grateful for the reviewer's excellent suggestion. Since PduCDE-sfGFP form an aggregate close to the cell pole in the absence of Pdu BMC induction (no 1,2-PD) (Fig. 3d), it is technically difficult to measure FRAP of PduCDE-sfGFP in the cytoplasm. Alternatively, another core enzyme PduGH-sfGFP without shell encapsulation showed dispersed distribution in the cytosol (Fig. 3d). Accordingly, we chose PduGH-sfGFP, which has inherent affinity for assembling with PduCDE to form the multi-complex enzymatic core (Fig. 3b and 3d), for comparison of the mobility of enzymes in solution versus "inside" the elongated Pdu BMC. We found that PduGH-sfGFP are highly dynamic in solution, and the fluorescence intensity recovered rapidly after only ~3 seconds, much faster than the dynamics of cargos within the Pdu BMC. We have added these data in Fig. S14 and Table S4.

d. Lines 160-170: Although the authors briefly reference structural defects (or absence of a shell) as a possible explanation of the growth enhancement results of Fig. 2D, I believe the authors should further clarify their explanation of the improved growth rate in strains with shell protein deletions. By "substrate availability" are the authors referring to the B12 cofactor, as I believe was used to rationalize the similar growth enhancements observed in the cited works? If so, the use of 'permeability' in this paragraph is vague and potentially misleading, as the permeability being measured is likely not to the substrate (1,2 PD) (or intermediates/products) of the metabolic segment inside the BMC. Instead, it seems the changes in 'permeability' that lead to an increased growth rate are being explained as the result of substantial structural compromise to the shell that abolishes its ability to act as a diffusive barrier because it is no longer intact, therefore allowing access of apo-PduCDE to the large B12 molecule that would otherwise be limiting to enzymes encapsulated inside a contiguous shell; however if this is the case, why are no deleterious effects of aldehyde leakage also observed for these deletions?

Reply: It is generally accepted that the Pdu BMC shell acts as a physical barrier to control the influx/efflux of substrates and cofactors. PduCDE is an Ado-B₁₂-dependent diol dehydratase, and the PduCDE activity is dependent on the B₁₂ level. The growth assays shown in the manuscript were performed at a limiting vitamin B₁₂ level. The absence or defect of the shell structure caused by deletion of shell proteins could elevate the availability of the large molecule vitamin B₁₂ in the Pdu BMC, thereby increasing the activities of encapsulated PduCDE.

However, the increase in the PduCDE activity is limited by the vitamin B₁₂ level, and the production of the toxic intermediate propionaldehyde is likely retained at a moderate level at a concentration that cells can survive. This would explain why the mutants grew faster than the WT at a limiting vitamin B₁₂ level.

In contrast, at a saturating vitamin B₁₂ level, the shell is no longer required to be a physical barrier due to the availability of vitamin B₁₂, and the PduCDE activity is maximized, resulting in enhanced propionaldehyde production. Shell encapsulation ensures sequestration of toxic intermediates and the fast growth of the WT cells, whereas the mutants with defective or lacking shell structures showed growth arrest between 18 and 32 h (Fig. S6). The same phenotypes at limiting and saturating levels of vitamin B₁₂ have also been reported previously (Cheng et al., *J. Bacteriol.* 2011, 193: 1385-1392).

To clarify this point, we have conducted new growth experiments at a saturating vitamin B₁₂ level (shown in Fig. S6) and have added relevant explanations in Results and Methods in the revised manuscript. We have altered the way the term 'permeability' is used in the revised manuscript.

e. In observing the growth rates of the Pdu mutants compared to the WT, results showed increased growth rates for most of the mutants. This is suggested to be due to formation of the enzymatic core

without encapsulation, allowing for greater access to environmental B₁₂. This seems the likely case but wouldn't this also release cytotoxic intermediates that would damage the cell? Would there be any way to quantify increased concentrations of these intermediates and compare them to the WT?

Reply: As explained above, we have now performed cell growth assays on the WT and the $\Delta pduB^{1-37}$ and $\Delta pduM$ strains at a saturating level of vitamin B₁₂ (Fig. S6). The results showed that the mutants with defective or lacking shell structures exhibited a growth arrest period between 18 and 32 h at a saturating level of vitamin B₁₂. Previous studies showed that the growth arrest period of the gene deletion mutants corresponded to a spike in propionaldehyde levels, whereas the propionaldehyde production stayed at a low level in the WT cells (Cheng et al., *J. Bacteriol.* 2011, 193: 1385-1392). The precise quantification of the intermediates in the WT and mutant strains will be carried out in our future studies.

f. Supplementary data with alignments / conservation: the sequences are compared among very similar Salmonella strains so a high conservation of amino acids is expected; are the pdu-associated sequences more conserved than expected for such similar strains?

Reply: As we responded to Reviewer 1 above, we have expanded the bioinformatic survey to other genera, including *Citrobacter*, *Escherichia*, and *Klebsiella*. We found that the sequences and structures of PduB/M/K were conserved among these species. The new results are shown in Fig. S10 in the revised manuscript.

g. potential effects of fluorescent labeling technique are not discussed (e.g. fluorescently tagged proteins might be insoluble and form inclusion bodies at poles)

Reply: To test the effects of fluorescent labelling technique, we designed and conducted a new series of control experiments in the absence and presence of 1,2-PD (Fig. 3b and 3d, Figs S2, S3). Amongst all the fluorescent tagged Pdu proteins, only PduCDE-sfGFP, PduP-mCherry, and PduO-mCherry formed aggregates at poles in the absence of 1,2-PD.

The aggregation of PduCDE-sfGFP and PduP-mCherry is likely due to their encapsulation peptides (the N-terminus of PduD and PduP) (Lee, Matthew J., et al. *Metab. Eng.* 2016, 36: 48-56) instead of fluorescent labelling. The aggregation of PduO-mCherry could result from self-aggregation, insolubility or the fusion with the fluorescence tag. We have added a discussion about the possible effects of fluorescence tagging, and have modified the text relevant to PduO accordingly in Results.

Minor comments

a. Lines 136-137: the authors should provide fluorescence images for the WT cells expressing PduJ-sfGFP and PduE-mCherry as a control for the $\Delta PduA$ strain images in Fig2A.

Reply: The fluorescence image of the WT cells expressing PduJ-sfGFP and PduE-mCherry as a control has been shown in Fig. S3 in the submitted manuscript.

b. Lines 142-143: to avoid confusion, it would be best to qualify (as done a few lines above) that PduJ-sfGFP was used for the $\Delta PduA$ strain.

Reply: We have clarified that PduJ-sfGFP was used for the $\Delta pduA$ strain in the revised manuscript.

c. Line 182: authors should specify that this could also be the reasonable result of fusion with the fluorophore.

Reply: We have modified the relevant text accordingly.

d. It wasn't initially clear that the PduCDE enzyme complexes in Fig3D were expressed from a plasmid and which subunit was tagged with sfGFP (although my assumption is PduE). I suggest adding this detail in the figure legend to avoid any confusion. Related to this point, the main text referring to this panel would also benefit from some elaboration that makes clear the difference between the fluorescence phenotypes observed here when PduCDE, with a fluorophore tagged

subunit, is expressed from a plasmid (if my interpretation is correct) as opposed to only one of the subunits (i.e. PduE-sfGFP shown in previous figures) that lack the PduC and PduD partners encoded only in the genome and not expressed due to the lack of 1,2-PD.

Reply: PduCDE enzyme complexes in Fig. 3d were expressed from a plasmid and the subunit PduE was tagged with sfGFP. We have included more details to clarify this important point in the figure legend of Fig. 3d and in the main text.

e. Fig5C “C” label overlapping with figure panel. Fig 5E: “pole” and “Midcell” labels overlapping with boundary box.

Reply: Thanks for pointing these out. We have modified the labels in Fig. 5 accordingly.

f. Lines 455-457: Authors refer to the PDU encapsulation peptides as a “model system of the assessment of the LLPS-mediated ordered assembly of protein organelles”, however the authors should clarify that encapsulation peptides may not meet the definition of conventional LLPS-mediated systems, as they have defined them, in part, by their secondary structure – an amphipathic alpha helix. References to EP literature and clarification/discussion of how EPs and characterized LLPS systems differ is warranted.

Reply: We have clarified that the secondary structure of the encapsulation peptides may not meet the definition of conventional LLPS. Relevant literature has also been cited.

g. Lines 488-490: authors should describe how these small volume (100 ul) cultures were grown and OD’s recorded – was it using the same plate reader system described later?

Reply: We have modified the text as follows: ‘Unless otherwise specified, 1 μ L of this culture was sub-inoculated to 100 μ L of MIM in 2 mL Eppendorf tube both in the absence and in the presence of 1,2-PD, shaken aerobically at 37°C for 10 h until OD₆₀₀ reaching 1.0 to 1.2 (DS-11 Spectrophotometer/Fluorometer Series from DeNovix)’.

h. Line 527: reference for the Gibson assembly method should be added: Daniel G. Gibson et al., “Enzymatic Assembly of DNA Molecules up to Several Hundred Kilobases,” Nature Methods 6, no. 5 (May 2009): 343–45, <https://doi.org/10.1038/nmeth.1318>.

Reply: We have added the reference as suggested.

i. In the Pearson’s R plots in the SI material, not all lines extend from the origin (e.g. S17 PduU-PduA), nor do all scatter plots appear to be symmetrical about the line. For example, Fig. S10 PduP-PduE is calculated as having a higher coefficient but doesn’t appear to be a better “fit” to the white line than the PduK-PduA. How do these examples impact interpretation of “colocalization”?

Reply: Scatterplots were generated by plotting the intensity value of each pixel of mCherry along the x-axis and the intensity value of the same pixel location of sfGFP on the y-axis using the Coloc2 plugins in ImageJ. The lines were fitted automatically by the software to represent the linear relationship between the fluorescent signals. It is not absolute that all lines should extend from the origin (Bolte and Fabrice P. *J. Microsc.* 2006, 224(3): 213-232).

In Fig S10B (now Fig. S11b in the revised manuscript), the dots on the diagram of PduK-PduA appear more symmetrical about the line, but the range of the cloud is slightly dispersed from the line. This observation could possibly explain why the calculated Pearson’s R of PduP-PduE is slightly higher than PduK-PduA.

line 236: Should those refer to Figure 4 instead of 2?

Reply: Here, we referred to Fig. 2 instead of Fig. 4, as Fig. 2 shows the correlation between PduE-mCherry and PduA-sfGFP, which allowed us to identify the importance of PduM. Based on the

finding in Fig. 2, we conducted extensive studies using various combinations of fluorescence tags in Fig. 4.

line 368: grammar, "assembly" should be "assemble"

Reply: We have changed "assembly" to "assemble".

line 934 / Figure 1: PduV is missing in list of different components, PduM is listed as shell protein but doesn't have a 00936/03319 pfam

Reply: We have modified the figure legend of Fig. 1 accordingly.

line 958 / Figure 2: error for the delta pduA measurement seems higher than the 0.02 when comparing to errors of other plots

Reply: Thanks for picking up the typo. We have corrected the data for $\Delta pduA$ to "0.12" instead of "0.02".

line 966 / Figure 3: panel H: make it clear that those two images are not from the same strain, e.g. deltaPduB1-37 vs deltaPduB1-37 WITH expression of fluorescently labeled proteins

Reply: We have labelled two images accordingly, and also clarified this in the figure legend.

line 1000 / Figure 4: panel H: as with Figure 3, those strains differ since the TEM image is not from the strain with fluorescently marked proteins

Reply: We have labelled two images accordingly, and also clarified this in the figure legend.

Reviewer #3:

The study presented by Liu group provides an in-depth study, where they have individually characterized the protein-protein interaction and assembly involved in catabolic bacterial microcompartment formation. Liu group in several of their earlier studies successfully utilized microscopy as an effective tool to study the complex traits of microcompartment formation. In this study, using fluorescent labeled individual BMC shell protein and an internal protein they answer one simple question “Which one happens first: the egg or the chicken?” or in the context of the study, whether the core forms first or the shell. While several earlier studies specifically from Bobik group has forwarded substantial knowledge (as also presented here), finally allowed this study to use all these fragmented knowledges into one single study.

However, I am a bit lost in too many microscopic pictures and where to find the real question. While I can understand one key point in this study is the role of PduM, this protein however is not present in all BMCs (reported till date), hence the function of PduM cannot be universal. The study thus is not focused and may not appeal to a broad audience for a journal like *Nar. Comm.*

Reply: We respectfully disagree with the reviewer on this point. Our revised bioinformatic analysis, shown in Fig. S10 in the revised manuscript, reveals that the three key proteins studied in this work, PduB¹⁻³⁷, PduM, and PduK, are present and conserved in a wide range of bacterial genera, including *Klebsiella*, *Escherichia*, *Citrobacter*, and *Salmonella*. Therefore, our study does provide insight into the general roles of PduB¹⁻³⁷, PduM, and PduK in mediating Pdu BMC assembly and subcellular distribution in diverse bacterial organisms.

Nonetheless, the study is interesting to people working in this field but needs to be improved:

1. Line 17: At the onset calling BMCs as organelles might not be correct.

Reply: The definition of organelle was coined Greening and Lithgow (*Nature Review Microbiology* 2020) as follows: “An organelle is a subcellular structure containing a proteomically-distinct interior and a defined boundary layer (whether lipid membrane, lipid monolayer, proteinaceous or phase-defined) that affects cellular physiology.” BMCs fit perfectly this organelle definition. Hence, the term organelle has been increasingly used in the BMC-related publications. See also recent review papers (Kerfeld et al, *Nat. Rev. Microbiol.* 2018, 16: 277-290; Liu et al, *Curr Opin Microbiol* 2021, 63: 133-141; Liu, *Microbial Biotechnol*, 2021, 14: 88-93).

2. Line 30: Which kind of cells?

Reply: Metabolic compartmentalization is ubiquitous in eukaryotic and prokaryotic cells. We have clarified this in the revised manuscript.

3. Line 47: which are examples of bacteria in anoxic niches? All those cited are only reviews.

Reply: *Salmonella* is a well-studied example of bacterium that inhabits in anoxic niches in the gastrointestinal tract. We have added relevant references to clarify this important point.

4. Line 127: The microscopy was done 1hr post induction with 1,2-PD, how do the cells look at the mid exponential phase?

Reply: For Pdu BMC birth event detection, the microscopy was done at 1 h post induction with 1,2-PD (Fig. 5). Other microscopic experiments were carried out following 10 h induction, until the OD₆₀₀ reached 1.0 to 1.2, the same growth phase that was used for Pdu BMC isolation in the present work and in previous studies (Sinha, et al, *J. Bacteriol.* 2012, 194: 1912-1918). We have added these details in Methods.

5. Line 141: the signals are in discrete patches; do they remain the same after 15 h (~ 0.5 OD)?

Reply: As explained in Comment 4 above, this signal was detected at 10 h (OD = 1.0-1.2).

6. In Figure 2B: $\Delta pduM$, in this strain PduB is in full length still PduE m-cherry do not co-localize? Is there a specific reason?

Reply: Yes, one of our key findings is that both PduB¹⁻³⁷ and PduM play multiple roles, being important for shell-interior interaction. PduB¹⁻³⁷ interacts with the internal core with the assistance of PduM. The absence of PduM causes the separation of cargo and shell assemblies. Please see details in the Results section of “PduM plays a role in binding between shell and enzyme core” and the Discussion section in the manuscript, as well as the model shown in Fig. 7.

7. Line 147-150: PduJ is not absolutely necessary for Pdu BMC formation since there are bacteria that do not have PduJ or this must only be in case of *S. enterica*. For the same reason, it is important to see the fluorescent signals within the cells at a later time point.

Reply: We agree with the reviewer that PduJ and PduA may be functionally redundant. However, PduJ is highly conserved in a wide range of bacterial genera, including *Salmonella*, *Citrobacter*, *Escherichia*, and *Klebsiella*. Please see our new bioinformatic analysis shown in Fig. S10 in the revised manuscript. As we explained in the response to Comment 4 above, the fluorescence signal was detected at 10 h (~ 1.0-1.2 OD).

8. Line 157: In $\Delta pduB^{1-37}$, if the assembly of shell and cargo (PduE) is spatially separated, how do the cells show higher growth phenotype? In the bioenergetic point of view, a faster growth phenotype is in direct correlation to its ATP turnover. Since the cells grow solely on 1,2-PD, propionate formation is likely the only energy conserving mechanism. A spatially separated PduCDE assembly outside BMC likely results propionaldehyde to be generated outside BMC which is supposed to be toxic to the cells. In the same argument, if we consider lowering VitB12 concentration results in the phenomenon, bioenergetically this is not possible (Fig 2D).

Reply: Please see the reply to Reviewer #2's Major Point 'b' above.

9. Line 181-182: The insoluble theory might not be true: it is likely PduO and PduP is constitutively expressed.

Reply: The expression of endogenous *pdu* operon (*pduA-X*) in the *Salmonella* genome is controlled by an inducible promoter that is activated by 1,2-PD and the Crp and Arc systems (Ailion, M., et al. *J. Bacteriol.* 1993, 175(22): 7200-7208; Chen et al., *J. Bacteriol.* 1995, 177(19): 5401-5410). In our established system, PduO and PduP were not constitutively expressed. All fluorescently tagged Pdu proteins in this manuscript were expressed from a pBAD plasmid. The fluorescence signal shown in Fig. 3b was detected in the absence of 1,2-PD, when all endogenous Pdu proteins (including PduO and PduP) were not expressed. Comparison of the fluorescence signals of PduE/PduG/PduL/PduQ with those of PduO and PduP shows that PduO and PduP were either insoluble or self-aggregated (Fig. 3).

10. If we take figure 2B $\Delta pduB^{1-37}$ and correlate figure 3B, this undoubtedly shows the BMC and the internal core protein localization is completely aloof. So, in this case, purification of BMC will result in purified shell, Is this true?

Reply: This is true. Our thin-section EM data of the $\Delta pduB^{1-37}$ cells confirmed formation of empty shell structures and a polar cargo aggregation that are spatially separated in the cells (Fig. 3h), consistent with our confocal observations and a previous study that reported empty shell structures in the purification of $\Delta pduB^{1-37}$ Pdu BMC (Lehman, et al. *J. Bacteriol.* 2017, 199: e00785-16). We have emphasised this point in the manuscript as follows:

“These results demonstrate that the shell proteins can self-associate to form multi-complex structures in the $\Delta pduB^{1-37}$ strain, which resemble the empty shell-like structures observed previously. We corroborated the fluorescence data with thin-section EM, which identified a polar aggregate (likely the core structure) and several shell-like particles in the $\Delta pduB^{1-37}$ cell (Fig. 3h).”

11. Line 238, 248-249: This might be true only with *Salmonella*, not all BMC clusters have PduM. In our experience, PduM is not absolute and intact BMCs are well formed without it.

Reply: Our manuscript has provided strong experimental evidence that PduM is required for cargo-shell assembly and Pdu BMC formation (Figs. 2, 4, 7, and relevant statements in the manuscript). Our new comparative genomic and structural prediction analysis has confirmed that the sequence and structure of PduM are conserved between the four Pdu BMC-containing bacterial genera: *Salmonella*, *Citrobacter*, *Escherichia*, and *Klebsiella* (Fig. S10), suggesting that PduM plays an important generic role in Pdu BMC biogenesis.

12. Line 265: in $\Delta pduB1-37/\Delta pduK$, Fig. 4F shows PduB' not to colocalize with PduA rather to be dispersed over the cell, am I correct?

Reply: As we have explained in the manuscript, PduB' in $\Delta pduB^{1-37}/\Delta pduK$ mainly formed polar aggregates, which were explicitly colocalized with PduA (Fig. 4f and Fig. R4 below). The cytosolic distribution of PduB' was also seen in the WT background (Fig. S3 in the manuscript), indicating that a portion of PduB' could not be integrated into Pdu BMC structures.

Fig. R4. Confocal images of the $\Delta pduB^{1-37}-\Delta pduK$ mutant expressing PduB' mCherry/PduA-sfGFP grown in MIM+1,2-PD media.

13. Fig 5. I am not an expert in fluorescence microscopy and hence cannot interpret most of the observed results and how relevant is it in current context. The results are dependent on several factors like style of experiment, time at which the results recorded. I am also not sure how tagging a GFP (27kda) protein to PduA (9kda) protein allows BMC formation (as I can gauge a 9kda protein will not interact similarly to its partner once it is modified to a 37kda protein).

Reply: In Fig. 5, the confocal microscopy was done at 1 h post induction with 1,2-PD for Pdu BMC birth event detection. This figure shows that shell and cargo of Pdu BMC assemble independently at cell poles. All experimental details (e.g. temperature and time) were provided in Methods.

Although we cannot exclude the possibility that tagging a GFP to shell proteins may interfere with BMC formation, fluorescence tagging has been extensively used in several key publications for the study of the assembly, stoichiometry, and dynamics of BMCs, including carboxysomes and Pdu BMCs (Cameron, Jeffrey C., et al. *Cell* 2013, 155: 1131-1140; Sun et al, *Plant Physiol*, 2016, 171(1): 530-541; Huang et al, *Plant Physiol*, 2019, 179(1): 184-194; Huang et al, *PNAS*, 2020, 117(29): 17418-17428; Yang et al, *Nat. Commun*, 2020, 11(1): 1-11; Held et al, *Sci. Rep*, 2016, 6(1): 1-15).

Here, to minimise interference between Pdu proteins and GFP, we used the “superfolder” version of GFP to tag proteins. Tagging with superfolder GFP reduces steric hindrance, and the fluorescently tagged Pdu proteins were expressed from a pBAD plasmid instead of from the chromosome. Accordingly, the assembled Pdu BMCs consist of both endogenous Pdu proteins and plasmid-expressed fluorescently-tagged Pdu proteins. Moreover, fluorescently-tagged Pdu proteins were produced in the absence of arabinose, driven by the low-level background expression of the pBAD promoter. Collectively, the low levels of fluorescently tagged Pdu proteins incorporated into the resulting Pdu BMCs are expected to have limited effects on the assembly and structural integrity of the Pdu BMCs.

14. Line 320-322: figure 3B actually shows core and BMC are separate as in my earlier comment. If core is separate, how can it be within BMC?

Reply: We have changed it to "of native Pdu BMCs" to clarify this sentence.

15. Fig. 6b: if I am not wrong: PduE is supposed to have an N-terminal encapsulation peptide and hence should localize similar to PduL or at least encapsulated within BMC (as per fluorescence signal is considered). Not having a discrete signal hints an unfolded protein or something else.

Reply: PduCDE has three protein subunits. It is the PduD protein that has an N-terminal encapsulation peptide (EP), not PduE (Fan, et al. *J. Bacteriol.* 2011, 193: 5623-5628). The PduD N-terminus ensures that the PduCDE complex to be encapsulated within the Pdu BMCs. Without EP, native PduE alone cannot be encapsulated in the absence of PduD, and generates discrete signal.

16. Fig 6F: Don't know how to interpret the data.

Reply: In Fig. 6f, we used fluorescence recovery after photobleaching (FRAP) and detected the fluorescence intensity changes in the laser bleached regions (yellow boxes), which indicate the dynamics of building proteins in the Pdu BMC. As the WT Pdu BMC is only about 100 nm in diameter, FRAP on these particles are technically difficult. Therefore, the elongated BMCs in $\Delta pduN$ were used for FRAP and for the quantitative analysis of the movement of Pdu BMC interiors *in vivo*. The same strategy has been used previously to study the diffusion of carboxysome core assemblies (Chen, et al. *PLoS One* 2013, 8: e76127).

Fig. 6f shows a series of confocal images as a function of time captured in the FRAP experiments, allowing us to determine the fluorescence kinetics (Fig. 6g). These results revealed that internal enzymes of the Pdu BMC possess a higher dynamics than shell proteins.

Reviewer comments, second round review –

Reviewer #2 (Remarks to the Author):

The authors have adequately addressed our queries.

Reviewer #3 (Remarks to the Author):

My comments have been addressed appropriately and I have no further concerns.

Responses to the Editor and Reviewer's Comments

Reviewer #3 is not convinced by your explanation that the mutagenesis method you apply strictly avoids polar effects and asks to genetically complement mutants in trans to rule out polar effects. Please revise your manuscript, addressing this remaining issue.

Response: To clarify how our approaches for generating gene-deletion mutants were designed to avoid polar effects and the sequence reads in the mutants to Reviewer #3, we have drawn a diagram of the *pdu* operon in the *Salmonella* Typhimurium LT2 WT and gene-deletion mutants (Fig. R1a, using $\Delta pduM$ as an example). In this study, we mainly used the Lambda Red method (Datsenko and Wanner, *PNAS*, 2000, 97:6640-6645) to perform gene deletion from the *Salmonella* LT2 genome. In each deletion, the coding sequence was replaced with an insertion of 84 bp including an FRT sequence (44 bp) and two primer sequences (20 bp each) (Fig. R1a). We have previously assessed the transcriptomes of 18 gene-deletion mutants of *Salmonella* Typhimurium LT2 generated by the Lambda Red approach (Fig. R1b) (Colgan et al., *PLoS Genetics* 2016, 12: e1006258). **The results demonstrated explicitly that the transcriptomes of these *S. Typhimurium* mutants were not affected by gene deletion using the Lambda Red approach, and the gene deletions did not cause significant polar downstream effects (Fig. R1b).**

Fig. R1. Diagram of the *pdu* operon in WT and gene-deletion mutants and transcriptome analysis of gene-deletion mutants by Lambda Red method. (a) An example of the diagram of the *pdu* operon in WT and *pduM*-deletion mutants using the Lambda Red and scarless approaches. In the $\Delta pduM$ mutant created by the Lambda Red approach, the coding sequence of *pduM* was replaced with an insertion of 84 bp including an FRT sequence (44 bp) and two primer sequences (20 bp each) that did not disrupt the reading frame or translational coupling. In the scarless $\Delta pduM$ mutant, the coding sequence of *pduM* was removed directly without any scar between the upstream *pduL* and downstream *pduN* genes. (b) An example of *Salmonella* chromosomal gene deletion (the *dam* gene that encodes the DNA adenine methyltransferase protein Dam) and transcriptome analysis (Colgan et al., *PLoS Genetics* 2016, 12(8): e1006258). The colors of each track represent the sequencing reads which map to that locus and the height of the normalized reads is directly proportional to the level of expression at that locus. The results showed clearly that neighboring genes were not affected by polar mutations. Black bent arrows indicate the transcriptional start sites (TSS).

To add independent confirmation of our findings, we generated 'scarless' *pduM* and *pduJ* deletion mutants by **genome editing** (Martínez-García and de Lorenzo, *Environ. Microbiol.* 2011, 13:2702-2716). Compared with the Lambda Red method, **genome editing does not introduce the FRT sequences at the gene deletion sites (Fig. R1a), diminishing the potential polar effects caused by gene deletion.** Confocal imaging showed similar

fluorescence profiles in the scarless mutants compared to those in the corresponding gene-deletion mutants generated by the Lambda Red method (Fig. R2). Collectively, our results confirmed that both Lambda Red and genome editing methods used in this work did not disrupt the reading frame (verified by DNA sequencing) or translational coupling, and thus did not cause polar effects.

Fig. R2. Confocal images of the $\Delta pduM/\Delta pduJ$ mutants and the scarless $\Delta pduM/\Delta pduJ$ mutants expressing PduE-mCherry/PduA-sfGFP grown in MIM+1,2-PD media.

To rule out the possibility of polar effects, our previously submitted manuscript has shown the complementation results of the $\Delta pduB^{1-37}$, $\Delta pduK$, and $\Delta pduM$ mutants by expressing PduB/PduK/PduM using a pXG10 plasmid (Fig. S9). In this revision, we further carried out complementation experiments of the $\Delta pduJ$ mutant, including growth assays and confocal imaging (Fig. R3, Fig. S6 in the revised manuscript). Full complementation was observed in the $\Delta pduB^{1-37}$ and $\Delta pduK$ mutants, as revealed by confocal, thin-section EM, and growth assays (Fig. S10 in the revised manuscript), indicating that the observed phenotypes were owing to the corresponding mutations, instead of polar effects on the expression of downstream *pdu* genes. Moreover, we have successfully isolated intact Pdu BMCs from the $\Delta pduK$ mutant, as confirmed by SDS-PAGE and EM (Fig. S12), also confirming no polar effects on the expression of downstream *pduP/S/Q/O/U* genes caused by gene deletion. For the $\Delta pduM$ and $\Delta pduJ$ mutants, growth assays showed that their phenotypes were well recovered by expressing PduM/PduJ in the $\Delta pduM/\Delta pduJ$ strain (Fig. S6 and S7 in the revised manuscript). In addition, the $\Delta pduM$ and $\Delta pduJ$ mutants grew well when using 1,2-PD as the sole carbon source, which requires proper expression of multiple genes that are downstream of *pduM/pduJ* (Fig. R3). Therefore, our results indicated that the $\Delta pduM$ and $\Delta pduJ$ mutants, together with $\Delta pduB^{1-37}$ and $\Delta pduK$ mutants, should be non-polar.

Fig. R3. Full complementation of PduB and PduK, and partial complementation of PduJ and PduM. Fluorescence images on LT2-WT and *pdu* gene-deletion mutants ($\Delta pduB^{1-37}$, $\Delta pduJ$, $\Delta pduK$, $\Delta pduM$) expressing PduE mCherry/PduA-sfGFP, and mutants that express corresponding proteins from a pXG10 plasmid grown in MIM+1,2-PD media (complementation experiments).

The fluorescence profiles of the $\Delta pduM/\Delta pduJ$ complementation strains showed some aberrant structures compared to that of the WT, reflecting partial complementation. We have recently revealed that the native Pdu BMC contains a low content of PduM (~50 per Pdu BMC) and a great content of PduJ (~870 per Pdu BMC) (Yang *et al.*, *Nat Commun*, 2020, 11:1976). The lack of full complementation in the $\Delta pduM$ and $\Delta pduJ$ mutants was probably due to the difficulty in gaining accurate expression levels of PduM and PduJ to ensure precise Pdu BMC assembly, as discussed in the manuscript. The partial complementation has also been reported in previous studies on Pdu BMCs (Cheng *et al.*, *J. Bacteriol.* 2011, 193:1385-1392), which stated as follows:

*"Three different pduA deletion mutants were constructed by linear transformation of PCR products, which is designed to generate nonpolar mutations, and **all three showed partial complementation**... It seems most likely to us that the **lack of full complementation was due to difficulty in establishing the precise gene dosage**. Therefore, despite the lack of full complementation, it is likely the phenotypes observed for the pduA deletion mutant resulting from the pduA deletion."*

Taken together, all our data demonstrated that the observed phenotypes were resulted from the corresponding gene deletions, and the possibility of polar effects caused by gene deletion can be ruled out. We have added new PduJ complementation results (Fig. S6) and made relevant changes in the revised manuscript.